# Improving GOES Advanced Baseline Imager (ABI) Aerosol Optical Depth (AOD) Retrievals using an Empirical Bias Correction Algorithm

Hai Zhang[1], Shobha Kondragunta[2,] Istvan Laszlo[2], Mi Zhou[1]

[1]I.M. Systems Group, 5825 University Research Ct, Suite 3250, College Park, MD 20740, USA.
[2]NOAA/NESDIS, 5825 University Research Ct, Suite 3250, College Park, MD 20740, USA.

*Correspondence to*: Hai Zhang (hai.zhang@noaa.gov)

**Abstract.** The Advanced Baseline Imager (ABI) on board the Geostationary Operational Environmental Satellite-R (GOES-R) series enables retrieval of aerosol optical depth (AOD) from geostationary satellites using a multi-band algorithm similar to those of polar-orbiting satellites' sensors, such as the Moderate Resolution Imaging Spectroradiometer (MODIS) and Visible Infrared Imaging Radiometer Suite (VIIRS). However, this work demonstrates that the current version of GOES-16 (GOES-East) ABI AOD has diurnally varying biases due to limitations in the land surface reflectance relationships between the 0.47 µm band and the 2.2 µm band and between 0.64 µm band and 2.2 µm band used in the ABI AOD retrieval algorithm, which vary with the Sun-satellite geometry and NDVI (Normalized Difference Vegetation Index). To reduce these biases, an empirical bias correction algorithm has been developed based on the lowest observed ABI AOD of an adjacent 30-day period and the background AOD at each time step and at each pixel. The bias correction algorithm improves the performance of ABI AOD compared to AErosol RObotic NETwork (AERONET) AOD, especially for the high and medium (top 2) quality ABI AOD. AOD data for the period August 6 to December 31, 2018 are used to evaluate the bias correction algorithm. After bias correction, the correlation between the top 2 quality ABI AOD and AERONET AOD improves from 0.87 to 0.91, the mean bias improves from 0.04 to 0.00, and root mean square error (RMSE) improves from 0.09 to 0.05. These results for the bias corrected top 2 qualities ABI AOD are comparable to those of the corrected high-quality ABI AOD. By using the top 2 qualities of ABI AOD in conjunction with the bias correction algorithm, the areal coverage of ABI AOD is increased by about 100% without loss of data accuracy.

## 1 Introduction

Aerosols in the atmosphere such as dust, smoke, pollutants, volcanic ash, and sea spray can affect climate through scattering and absorption of radiation directly, and through interaction with clouds indirectly (Albrecht, 1989; Rosenfeld and Lensky, 1998; Mahowald, 2011). In addition, aerosols impact air quality and thus affect human health (e.g. Pope and Dockery 2006). Satellite retrieved aerosol optical depth (AOD), a quantitative measure of the amount of aerosols present in the atmosphere, is

useful for evaluating aerosols' effect on climate change (e.g. Yu et al. 2006) and for estimating and forecasting ambient $PM_{2.5}$ concentrations (particulate matter with median diameter $\leq 2.5$ μm; e.g. Hoff and Christopher, 2009).

AOD from polar-orbiting satellite sensors, such as the Moderate Resolution Imaging Spectroradiometer (MODIS) and Visible Infrared Imaging Radiometer Suite (VIIRS), is retrieved using multi-channel algorithms (Levy et al., 2007; Levy et al. 2010;

Sayer et al., 2014; Jackson et al., 2013; Liu et al., 2014; Laszlo and Liu, 2016). As a result, AOD from MODIS and VIIRS has high accuracy, e.g. MODIS dark target AOD has an expected error of ±(0.05 +15 %) over land (Levy et al. 2013) and VIIRS AOD developed at the National Oceanic and Atmospheric Administration (NOAA) has a bias of 0.02 and standard deviation of error of 0.11 (Laszlo and Liu, 2016), but the low temporal resolution of polar-orbiting satellites limits the availability of observations for a given location. In contrast, geostationary satellites such as the United States' Geostationary

Operational Environmental Satellites (GOES) provide an opportunity for nearly continuous AOD retrievals during daylight over a hemispheric domain. The NOAA GOES Aerosol and Smoke Product (GASP) retrieved from the legacy GOES imagers, however, was not as accurate as the MODIS or VIIRS AOD due to limitations imposed by a single channel retrieval (Prados et al., 2007; Green et al., 2009). GASP AOD was reported to have a correlation of 0.79 and RMSE of 0.13 compared with AERONET AOD over CONUS (Prados et al., 2007). The Advanced Baseline Imager (ABI) on the new generation GOES-R

series of satellites is expected to provide AOD retrievals with accuracies similar to those from MODIS and VIIRS due to similar instrument design and algorithm science, combined with high temporal resolution. NOAA launched the first and the second satellites in the GOES-R series, GOES-16 and GOES-17, in 2016 and 2018, respectively (Schmit et al., 2017; https://www.nesdis.noaa.gov/content/goes-17-now-operational-here%E2%80%99s-what-it-means-weather-forecasts-western-us accessed September 1, 2020). Each satellite carries an ABI, which has 16 spectral bands ranging from the visible

to infrared wavelengths. GOES-16 is located at 75.2°W and GOES-17 is located at 137.2 °W. Both satellites observe the continental United States (CONUS) region every 5 minutes and the full hemispheric disk every 10 minutes or every 15 minutes, depending on the scan mode (Schmit et al., 2017).

The NOAA ABI AOD product has a spatial resolution of 2 km at nadir, compared to 3 km and 10 km from MODIS Collection

6 and 750 m (NOAA product) and 6 km (NASA product) from VIIRS. The GOES-16 ABI AOD product was released on July 25, 2018, while the GOES-17 ABI AOD product reached provisional maturity on Jaunary 1, 2019; Definition for provisional maturity can be found in EOSDIS Glossary (https://earthdata.nasa.gov/learn/user-resources/glossary, accessed May 14, 2020).

The accuracy and precision of VIIRS and MODIS AOD is well documented for use in various decision support systems (Laszlo

and Liu, 2016; Sawyer et al., 2020; Levy et al., 2013; Sayer et al., 2014). The geometries of observations from a geostationary satellite are quite different from a polar-orbiting satellite; this can lead to differences in the quality of retrieved AOD despite the similarity of the AOD retrieval algorithms. It is therefore very important to evaluate the new ABI AOD product and

demonstrate its accuracy and precision at daily and sub-daily time scales. This should allow users to interpret the ABI AOD product correctly and apply it appropriately in research and operational applications.


In this study, we compare GOES-16 ABI AODs to AERONET AODs for a five-month period in 2018 and document a diurnal bias in the ABI AOD due to deficiencies in the land surface reflectance relationship currently applied in the retrieval algorithm. The presence of the bias is consistent across the CONUS but its magnitude varies by location. We describe a novel method that corrects the bias for each AOD pixel and time step. The resultant corrected ABI AOD shows little to no diurnal bias over

a variety of surface types (e.g., urban, rural).

## 2 Data

### 2.1 GOES-16 ABI AOD

The GOES-16 ABI AOD data used in this work is from the period of August 6 to December 31, 2018, over the CONUS region. The ABI AOD data have 2 km spatial resolution at nadir and 5 minutes temporal resolution. Similar to MODIS and VIIRS

AOD, ABI AOD are retrieved using separate algorithms over ocean and over land, due to the different surface characteristics of ocean and land (Kondragunta et al., 2020; GOES-R AOD ATBD, 2018). Over land, three ABI channels are used in the retrieval, i.e. 0.47 µm, 0.64 µm, and 2.2 µm. The algorithm assumes linear relationships exist between the surface reflectance of 0.47 µm band and 2.2 µm band, and between 0.64 µm band and 2.2 µm band. The coefficients of the relationships are functions of NDVI and solar zenith angle (GOES-R ABI AOD ATBD, 2018). NDVI is between 0.86 and 0.64 µm channel

and is defined in the following equation:

$$\text{NDVI} = \frac{\rho_{0.86}^{\text{TOA}} - \rho_{0.64}^{\text{TOA}}}{\rho_{0.86}^{\text{TOA}} + \rho_{0.64}^{\text{TOA}}} \quad , \tag{1}$$

where $\rho_{0.64}^{\text{TOA}}$ and $\rho_{0.86}^{\text{TOA}}$ are TOA reflectances at 0.64 µm and 0.86 µm channels respectively.

Other atmospheric and geographic parameters needed for the retrieval are also inputted, such as surface pressure, surface height, total column ozone, etc. The algorithm only retrieves AOD over dark surface, when the TOA reflectance in the 2.2 µm

band is less than 0.25. The retrieval algorithm contains two steps. In the first step, one of four aerosol models is assumed, i.e. dust, smoke, urban, and generic, and AOD for each of the aerosol model is retrieved using the 0.47 µm and the 2.2 µm bands. The algorithm uses a Look-up-Table (LUT) to perform radiative transfer calculation. The LUT stores reflectances, transmittances and other quantities for discrete states of atmosphere and Sun-satellite geometries. For each AOD in the LUT, the algorithm performs atmospheric correction in 2.2 µm band to obtain surface reflectance in that band, and uses the 0.47 µm

and the 2.2 µm band relationship to obtain the 0.47 µm band surface reflectance. TOA reflectance in the 0.47 µm band is then calculated using the LUT. The AOD for the assumed aerosol model is obtained through interpolation of the two AODs that give TOA reflectances in the 0.47 µm band closest to the satellite measurement. At the end of this step, there are four AOD solutions from the 0.47 µm band and 2.2 µm band, one for each aerosol model. In the second step, one of the four solutions

is selected as the final retrieval using the 0.64 µm channel by looking for the aerosol model that gives a TOA reflectance in that channel that is the closest to the observed TOA reflectance. The algorithm does not make retrievals over bright land pixels, pixels covered by cloud or snow, etc. The AOD retrieval range is [-0.05,5] and any retrievals greater than 5 are marked as out of range.

The retrieval algorithm assigns the pixel level AOD to one of three qualities: high, medium and low. AOD quality is determined on conditions of the pixels, such as solar/satellite zenith angle, cloud/shadow adjacency, standard deviation of measured reflectance at a specific band. The full set of criteria used for assigning a quality level is listed in Table 1. High quality AOD is the most accurate and the one recommended for scientific applications. However, the ABI AOD retrieval algorithm uses such strict criteria to remove potential erroneous pixels that the number of pixels with high quality AOD is usually very small. For example, the ratio between the number of the top 2 qualities and the high quality matchup with AERONET is about 2 (see the following section), while the ratio is 1.2 for NOAA VIIRS AOD (Laszlo and Liu, 2016). The following criteria are used to degrade a pixel from high quality to medium quality: (1) adjacent to a cloudy pixel; (2) adjacent to a snow pixel within 3 pixels distance; (3) 3x3 standard deviation of 2 km 0.47 µm TOA reflectance is greater than 0.006; (4) retrieval residual is greater than 0.4; (5) external cloud mask is "probably clear" (instead of "confidently clear"). Out of these five criteria, the standard deviation test tends to remove a large number of pixels that are potentially high quality, i.e. about 65-80% in medium quality land pixels have standard deviation in the 0.47 µm band above the threshold of 0.006. This test is used to remove pixels that are inhomogeneous in TOA reflectance due to the existence of undetected cloud or snow by the cloud mask algorithm. A similar test is used in the NOAA VIIRS AOD algorithm but with the 0.41 µm band instead of the 0.47 µm band (e.g. Huang et al., 2018). The surface reflectance in the 0.41 µm channel is usually low and therefore does not have much influence in the standard deviation at the TOA for NOAA VIIRS AOD. Over the CONUS region, from VIIRS data, the 0.41 µm surface reflectance is 0.3-0.4 times the 0.67 µm band surface reflectance and the 0.47 µm surface reflectance is 0.5-0.6 times the 0.67 µm surface reflectance (Zhang et al., 2016). Therefore, 0.41 µm surface reflectance is about 20%-50% lower than 0.47 µm surface reflectance. However, the ABI does not have a 0.41 µm channel and the algorithm has to use the 0.47 µm channel instead. The surface can have a noticeable influence on the standard deviation in the 0.47 µm channel, especially in urban regions where surface reflectance variations are large. To include more retrieval pixels that are otherwise omitted due to the very conservative screening process for high quality pixels, both high quality and medium quality pixels are included in this analysis.

The surface reflectance relationship used in the operational ABI AOD algorithm was derived from AERONET matchup dataset using strict criteria, with cloud screening using that for high quality, low AERONET AOD (<0.2), 5 km within AERONET, etc., in order to minimize the cloud and aerosol model interference. Details of the criteria can be found in ABI AOD ATBD (2018).

The NASA Dark Target (DT) aerosol algorithm team applied their DT algorithm on geostationary satellite data such as ABI and AHI (Advanced Himawari Imager, Gupta et al., 2019). In order to test the bias correction algorithm on other AOD retrieval

algorithms, the GOES-16 ABI DT AOD was obtained from NASA for the month of July 2019. The product covers the full disk with 10 minutes temporal resolution and 10 km pixel resolution (at nadir).

## 2.2 AERONET AOD

The AErosol RObotic NETwork (AERONET) is a global ground-based aerosol remote sensing network (Holben et al., 1998). It uses CIMEL sun photometers to measure spectral sun irradiance and sky radiances. The measurements are then used to

calculate and retrieve aerosol properties. Among them, AOD is one of the main products; it is measured at a subset of 22 different wavelengths from ultraviolet to infrared, i.e. 340, 380, 400, 412, 440, 443, 490, 500, 510, 531, 532, 551, 555, 560, 620, 667, 675, 779, 865, 870, 1020, and 1640 nm, depending on the specific instrument. Ångström Exponent (AE) can be calculated from the multispectral AOD. Besides AOD, AERONET also retrieves other aerosol properties, such as volume size distribution, refractive index, phase function, and single scattering albedo (SSA). AERONET AOD is considered ground truth

for satellite AOD (Holben et al., 1998) and is used to evaluate the ABI AOD retrievals. AERONET AOD at 550 nm is obtained through interpolation from other spectral bands so that it can be compared against ABI AOD, which is reported at 550 nm. In this work, AERONET AOD version 3 level 1.5 is used. Although level 2.0 data have higher quality, they have time delays such that the latest data were not available during the analysis period. Level 1.5 AERONET AOD data is cloud screened and quality controlled, with an up to + 0.02 bias and one sigma uncertainty of 0.02 (Giles et al., 2019).


## 3 GOES-16 ABI AOD Diurnal Bias

The diurnal bias of ABI AOD is evident when it is compared to coincident measurements of AERONET AOD. The diurnal bias is most apparent on "clear" days, when AERONET AOD is $\leq 0.05$ during an entire day. Comparisons are made on clear days at six AERONET sites, listed in Table 2. These sites include a range of geographic locations across the CONUS and

different surface types (e.g., urban, suburban, rural), most of which are urban or surfaces with little vegetation. Matchups at the AERONET sites were made by averaging ABI AOD pixels within a circle of 27.5-km radius surrounding the site; a minimum of 120 pixels are required to have an effective matchup, which is about 20% of all the pixels within the circle. These criteria are adopted from the traditional satellite and AERONET AOD matchup procedure developed and recommended by Ichoku et al.(2002)


To illustrate the problem of the diurnal bias of ABI AOD the time series of ABI AOD and AERONET AOD for clear days are plotted at the representative AERONET sites in Figure 1. As demonstrated in the figure, the number of the ABI top 2 qualities (high and medium quality) data points are much larger than that of the high quality AOD. For example, on October 18, 2018

at the CCNY site (Figure 1a), which is located in New York City, New York, no high quality ABI AOD data matchup data are available, but top 2 qualities AOD matchup points exist at nearly all time steps.

The diurnal variation of the ABI AOD bias is observed at all six sites, but the magnitude of the bias varies, with higher bias observed at the urban/suburban sites (Figures 1a, 1c, 1d, and 1e) compared to the rural sites (Figures 1b and 1f). For all sites, the bias peaks around 17:00 UTC, when the Sun moves from the east of the satellite to the west of the satellite, as determined by the location of the satellite, i.e. 75.2°W for GOES-16. The bias curves are nearly symmetric at the two sites with longitudes close to that of the satellite (Figures 1a, 1b, and 1c), while the bias curves are asymmetric at the sites to the west of the satellite (Figures 1d, 1e, and 1f).

There are several potential causes of the diurnal bias observed in ABI AOD, including known sources of uncertainty associated with calibration, cloud/snow contaminations, aerosol models, and errors in the surface reflectance model (Li, et al., 2009). In the cases shown in Figure 1, all days have low AOD values and continuous AOD measurements from AERONET, indicating that the influences of the aerosol model selection and cloud contamination are small. Snow contamination is not an issue because the analysis days are mostly in September and October, before it was cold enough for widespread snowfall. The one case in December (University of Houston) was not contaminated by snow through visual inspection of the true color (RGB) images of VIIRS or GOES, which are available on the AerosolWatch website (https://www.star.nesdis.noaa.gov/smcd/spb/aq/AerosolWatch/, accessed September 1, 2020) Therefore, we hypothesize that the most probable reason for the observed diurnal patterns of the ABI AOD biases is errors in surface reflectance retrievals. In the ABI AOD retrieval algorithm, the land surface reflectance relationships between the 0.47 µm and the 2.2 µm band and between the 0.64 µm and the 2.2 µm band were parameterized, as described in Section 2.1, and assumed to be functions of solar zenith angle and NDVI. Errors in these parameterizations are most likely responsible for the observed diurnal pattern of the ABI AOD biases. When the deviation of parameterization from the actual relationship is large, the AOD retrieval error will also be large. One reason that causes the land surface relation error is that current surface relationships were derived from the dataset when GOES-16 was located at the test position (89.5°W) instead of the current operational position (75.2 °W), and so the relationship does not adequately represent the current observation geometry. When the satellite position changed, the characteristics such as reflectances due to the change in geometry and type of the surfaces being observed are no longer similar. The other reason is that the relationships are derived from training pixels selected using more strict criteria and therefore the pixels with relaxed criteria such as medium quality pixels may not be represented well by the training set.

The diurnal pattern of biases is also found to be different on different days. As an example, Figure 2 shows the diurnal bias at GSFC on two additional days in October 2018, the 18th and the 30th. Although the peak of the bias occurs at approximately the same time on both days, around 17:00 UTC, the magnitudes of the peaks are different. On October 12th (Figure 1a) the maximum ABI AOD is about 0.25, while it is 0.2 on October 18th (Figure 2a) and only 0.1 on October 30th (Figure 2b).

To further illustrate the reasons that cause the diurnal variation of the ABI AOD biases, atmospheric corrections were performed to obtain the surface reflectance at different times and days for the pixels near GSFC site, i.e. at 17:02 UTC and 20:02 UTC on October 12th, October 18th, and October 30th. The atmospheric correction uses the LUT from the ABI AOD retrieval and the input of the TOA reflectance from ABI, geometries, and AERONET AOD, along with the assumptions of standard column ozone, water vapor and surface pressure. Because there are four aerosol models in the LUT, the four surface reflectance values were averaged. In the ABI AOD retrieval algorithm, 0.47 µm and 2.2 µm bands are used to obtain AOD and surface reflectance and the 0.64 µm band is used to select aerosol model. Therefore, in this analysis, only the surface reflectance of the 0.47 µm and the 2.2 µm bands are obtained to illustrate the problem. Figure 3 shows the scatter plots of surface reflectances at 0.47 µm vs 2.2 µm of the pixels (with high and medium AOD quality) for the six scenarios, along with the corresponding NDVI histograms.

In the scatter plots, the average of the three days' solar zenith angle is used to calculate the coefficients of the linear relationships for each time step for illustration purpose, because the solar zenith angles are close in value for the three days at each time step with about ±2° differences. Here only two lines are plotted because the majority of the pixels have NDVI in these two categories, as shown in Figure 3 (c) and (d).

At 17:02 UTC, on October 30th 2018, nearly all the pixels fall into the category of 0.3≤ NDVI < 0.55 and the corresponding relationship line (orange) passes through nearly the center of the pixel groups. Therefore, the AOD retrieval at this time on October 30th uses a relationship close to the actual one and the AOD retrieved is close to AERONET AOD. On the other two days, about half of the pixels fall into 0.3≤ NDVI < 0.55 and another half into NDVI ≥ 0.55. Although the pixels with 0.3≤ NDVI < 0.55 use the relationship close to the actual one, the pixels with NDVI ≥ 0.55 use a relation far away from reality and therefore the retrievals have a large bias, i.e. about 0.2. Of these two days, October 12th has more fraction of pixels in the category with wrong relationship and therefore it has a slightly higher bias.

Comparing the two time steps, pixels have lower NDVI at 20:02 UTC than those at 17:02 UTC on the same days. The surface reflectance is significantly lower at 20:02 UTC, i.e. with mean surface reflectance reduced from 0.06 to 0.04 in 0.47 µm band. Again, October 30th at 20:02 UTC, the pixels use surface reflectance relation of 0.3≤ NDVI < 0.55, which is also close to the correct one. Although the other two days also use both relationships, both relationships are closer to the reality than the one with NDVI≥0.55 at 17:02 UTC. Therefore, all three cases at 20:02 UTC have retrievals close to AERONET AOD.

The change in NDVI from October 12th, 18th to October 30th is most likely due to the change in the colors of the vegetation during fall, when the leaves of trees turn reddish. Within the same day, due to the change in geometry, NDVI changed. It should be pointed out that even though at 20:02 UTC the surface relationships used are close to reality, there is still a lot of

scatter in the individual pixels. This can introduce pixel level uncertainty which cannot be observed when averaged over the area around AERONET site.

## 4 Bias Correction Algorithm

Now that the source of the diurnal bias in ABI AOD has been identified, the next step is to develop an algorithm to correct it by taking advantage of the special characteristics of geostationary satellites. Because the GOES-16 satellite is stationary, the locations of the image pixels are fixed and the satellite zenith and azimuthal angles remain unchanged. In addition, the solar zenith and azimuthal angles at a given time of day change little during a relatively short time period (< one month). These features, common to geostationary satellites, were used to design an AOD retrieval algorithm for the legacy GOES, e.g. the

GOES aerosol/smoke product (GASP) (Knapp, 2002a; Knapp et al., 2002b; Knapp et al., 2005; Prados, et al., 2007). Unlike the GOES-R series satellites, the imager onboard the legacy GOES had only one visible channel that was used for AOD retrieval. In the GASP retrieval algorithm, to obtain the surface reflectance at the visible channel at each time step, a composite TOA reflectance was generated such that the second lowest reflectance was chosen from a time period of the previous 28 days. This reflectance was then used to retrieve the surface reflectance assuming a background AOD of 0.02.

    We designed a GOES-16 ABI AOD bias correction algorithm similar to the GASP AOD retrieval algorithm. However, instead of reflectance space, the composite bias correction algorithm works in AOD space. The basic idea to derive the ABI AOD bias is that the minimum of a month ABI AOD at each time step should be close to the background AOD. Therefore, deviation of the minimum of ABI AOD retrievals during the one-month period from the AERONET-derived background AOD are assumed

to represent a systematic bias. The AOD bias at higher AOD load is estimated to be the same as the one obtained at the background AOD, which will be proved in section 6 through radiative transfer simulation. The AOD bias can then be removed from the original ABI AOD by subtraction.

    The flowchart of the algorithm is shown in Figure 4. GOES-16 ABI AOD top 2 qualities, i.e. high quality and medium quality,

are used to generate the bias curves in the algorithm, because the criteria for high quality AOD are very conservative and the standard deviation test that moves data from high quality to medium quality is very stringent and throws away a lot of good retrievals. The top 2 qualities data have much larger area coverage than the high quality data alone. For example, it is not possible to build a bias curve for pixels near CCNY using high quality AOD data as there are too few data points, as seen in Figure 1.

    In the bias correction algorithm, ABI AOD (top 2 qualities) over the CONUS with 5 minutes temporal resolution is first aggregated into 15 minutes temporal resolution. This is because GOES can operate in different modes and the observation times are different for different modes, even though the time interval between the time steps stays the same for the CONUS

region. Averaging AOD into 15 minutes intervals reorganizes the AOD data into regular time steps. In addition, averaging
AOD also increases data coverage at each time step. At each time step, the algorithm loops through a 30-day period to look
for the lowest AOD for each pixel. In this work, the 30-day time period was selected based on Prados et al. (2007). For real-
time bias correction, the most recent past 30 days are used, because future AOD observations, after the date of interest, are not
yet available. If the bias correction is being done as part of reprocessing, such that all the AOD data after the date of interest
are available, a 30-day period is used with the date of interest placed at the center; this period may estimate the AOD bias more
accurately. As shown in Knapp et al. (2005), the optimal time period to obtain a clear day background is not fixed and is
dependent on seasons.

Once the optimum 30-day period has been selected, the bias at each pixel and at each time step is estimated using the lowest
AOD during the 30-day period minus the background AOD. The background AOD over the CONUS area is obtained through
an analysis of multi-year AERONET AOD data using the method described in Zhang et al. (2016). The main steps are
summarized here for reference. At each AERONET site $i$, the lowest 5[th] percentile of AOD over a 5-year (2012-2016) period
is obtained and is set as the estimate of the background AOD ($\tau_i$) at the site. Then the background AOD at each site is
interpolated to provide continuous values across the globe using:

$$\tau_b = \frac{\sum_i w_i \tau_i}{\sum_i w_i}, \tag{2}$$

where $\tau_b$ is the interpolated background AOD, and $\tau_i$ is the background AOD at site $i$. The weighting factor $w_i$ is defined as a
function of the distance ($d_i$) between the site $i$ and the interpolation point as:

$$w_i = exp(-d_i/d_0), \tag{3}$$

where the constant $d_0$ is set as 500 km. Using this method, a global map of background AOD is obtained. The background
AOD over the CONUS is found to be low and the variation is also small, i.e. the average background AOD over CONUS is
0.025 and the range is [0.019, 0.033]. Therefore, instead of using various background AOD values at different places in the
bias correction algorithm, a constant background AOD of 0.025 is used, which is similar in magnitude as that used in GASP
algorithm. After the bias at each 15-minute time step is obtained for each pixel, the bias data are fitted to two curves of
polynomial of second order, separated at 17:00 UTC, which is about the time when the bias peaks. This step is used to obtain
estimates of the bias at each 5-minute AOD observation time step and also helps to further smooth the diurnal curve of the
bias. The use of a smoothed curve removes potential random noise from factors such as cloud shadow contamination and
deviations from background AOD at the lowest AOD retrieval. Subsequently, the bias corrected AOD is calculated by
subtracting the bias at each pixel for each time step from the original AOD. Background AOD may change over time in case
some extreme events happen, in which the bias correction algorithm may not work well. In this case, overcorrection in AOD
is expected because the bias is overestimated.

An example of a 2-km pixel close to GSFC is shown in Figure 5, where AOD is plotted as a function of time for the 30-day period from September 12 to October 11, 2018. The AOD lower bound is derived from the time period and is shown as red curve. The bias is estimated using the lower bound minus the background AOD of 0.025. It is then subsequently used to correct the bias for that pixel for the day October 12, 2018.


## 5 Bias Correction Algorithm Validation

### 5.1 Application to NOAA ABI AOD Data


GOES-16 ABI AOD data and AERONET AOD data for the time period from August 6, 2018 to December 31, 2018 are used to validate the bias correction algorithm. The diurnal bias of ABI AOD data across the CONUS domain was corrected using the algorithm described in Section 4 and compared to coincident AERONET AOD. The original ABI AOD and the bias corrected ABI AOD were matched with AERONET AOD using the following criteria: (1) ABI AOD are averaged within the

circle of 27.5 km radius around an AERONET site, requiring at least 120 valid AOD pixels within the circle; (2) AERONET AOD are averaged within ±30 minutes of the satellite observation time and at least 2 AERONET AOD data points exist within the hour. These are the same criteria that were used to validate the NOAA VIIRS AOD product (Liu, et al., 2014; Huang et al., 2016).

For the first 30 days of the validation period (August 6 to September 4), the bias correction curves are derived from the same 30 day period. For the remainder of the validation period, the bias correction curves are derived from the 30-day period immediately prior to the day of interest.

Figure 6 shows scatter plots of GOES-16 ABI AOD vs AERONET AOD for high quality and top 2 qualities of ABI AOD,

before and after bias correction, averaged over the entire validation period and across the CONUS domain. Scatter plots for both high quality and top 2 qualities are shown, although the bias curves were derived using the top 2 qualities data. In order for a valid comparison, the AOD pixels in the plots have one-to-one correspondence before and after bias corrections, i.e. the quality flag does not change and all the pixels are kept even though some of them may be below the lower bound of the operational GOES-16 ABI AOD product (-0.05) after bias correction. As seen in the scatter plots, the bias correction improves

the performance of the top 2 qualities ABI AOD more than the high-quality ABI AOD, which indicates that the ABI AOD algorithm does a good job identifying high quality retrievals. Therefore, the ABI AOD retrieval algorithm does a good job

identifying high quality retrievals, but with limited data coverage compared to the top 2 qualities. For the top 2 qualities ABI AOD, after bias correction, the correlation between ABI AOD and AERONET AOD improves from 0.87 to 0.91, the total bias improves from 0.04 to 0.00, and RMSE improves from 0.09 to 0.05. The high-quality ABI AOD shows a small decrease in 
RMSE, which improves from 0.06 to 0.05 after bias correction. The results in Figure 6 demonstrate that by applying the simple bias correction, the top 2 qualities ABI AOD perform as well as the high-quality ABI AOD, but with twice the number of matchups. In this way, the spatial coverage of ABI AOD is substantially increased, without loss of data accuracy, by using top 2 qualities in conjunction with the bias correction.

Table 3 shows validation statistics for GOES-16 ABI AOD vs AERONET AOD at the 6 AERONET sites listed in Table 2. After applying the bias correction, most of the statistics for ABI AOD improve at the six sites, demonstrating the success of the bias correction algorithm. For example, 5 out of 6 sites have RMSE improved to 0.05 or below. The exception is the University of Houston site, where the RMSE is still as high as 0.08 after correction, although it is improved from 0.19. This result may indicate there is still some bias left uncorrected at this site due to its complicated surface with respect to geometries. 
The sites in the eastern US have a geometry symmetric to the local noon and therefore the AOD biases are symmetric to the local noon. The sites in the western US do not have such symmetry and therefore the splitting of parameterization at noon and using second order polynomials may introduce some errors. The complexity of surfaces over University of Houston can be seen in Figure 1 (e), where two AOD bias peaks are observed, one in the morning and the other at noon, indicating that the diurnal variation of surface reflectance relationship is different from the other sites, such as GSFC, CCNY, etc, where AOD 
biases only peak at noon.

Figure 7 demonstrates the scattering angle dependence of the ABI AOD errors for high quality and top 2 qualities. It can be seen that the errors before bias correction have strong scattering angle dependency: AODs have positive bias when the scattering angle greater than 110° and negative bias otherwise; The bias increases with scattering angle, with the highest bias 
at 175° bin; top 2 qualities AOD has higher bias than high quality AOD, as expected. The scattering angle dependence of AOD retrieval bias may be caused by many reasons, in which surface reflectance modeling error is one of the main reasons (She et al., 2019). After applying the bias correction, the positive biases in both high quality and top 2 qualities for scattering angle greater than 110° are removed. The standard deviations of the errors are also smaller in most of the bins. The bias correction does not have much improvements in bias for the scattering angle less than 110° as large as those greater than 110°. 

To evaluate the performance of the algorithm for a range of AODs, Figure 8 shows the ABI AOD error and standard deviation in different AERONET AOD bins, with equal number of matchup data in each bin. For high quality AOD, bias correction reduces bias in the highest two AOD bins, with center around 0.3 and 0.57. In the range [0.1, 0.3], bias correction over corrects and introduces negative mean bias with slightly larger magnitude than the original mean bias, around 0.01 in magnitude 
differences. In the range [0,0.1], AOD mean biases are close to zero both before and after correction, but the bias correction

AOD error has smaller standard deviation. For the top 2 qualities ABI AOD, bias correction reduces the bias for all ranges of AODs with slight over corrections of magnitude of about 0.02 when AOD is greater than 0.1.

Figure 9 shows the monthly mean AOD for September 2018 at three time steps, i.e. 1500 UTC, 1700 UTC and 2000 UTC. At each time step AOD is first composited within ±30 minutes and then averaged over the month. A pixel has an effective mean AOD if there are at least six days with AOD retrievals with high or medium quality. The observed diurnal pattern across CONUS is similar to examples shown in Figure 1 for some AERONET sites. The morning 1500 UTC and the afternoon 2000 UTC mean AOD have lower values than that at noon 1700 UTC. After the bias correction, the three time steps have closer mean AOD, which is expected. By comparing the figures between the original and bias corrected AOD map, one can see a lot of places have AOD biases of about 0.1 to 0.2. The biases are higher at noon than in the morning and in the afternoon. These maps also demonstrate that the AOD biases exist not only at the AERONET examples shown in the previous sections but also in most of the places across the domain.

Figure 10 shows the maps of the statistical metrics over AERONET sites with more than 400 matchups for the correlation coefficients, mean biases and RMSEs for the original ABI AOD (top 2 qualities) vs AERONET AOD and for the bias corrected ABI AOD (top 2 qualities) vs AERONET AOD. As can be seen, over most of the sites, the performances of bias corrected AOD improve compared to the original AODs. In the original ABI AOD, no geographical pattern of the performances is observed. Especially noteworthy is that AOD retrievals for some sites that are very close to each other have very difference performance metrics. There are no AERONET matchup in the western US, because the ABI AOD restrict the satellite view zenith angle to those below 60°. The western US usually have heavy smokes due to wild fires.

Most of the sites with high bias (around 0.1 or above) and RMSE (around 0.15 or above) before bias correction are urban sites. For example, Tucson, University of Houston, CCNY, which have already been shown in previous analysis. There are two sites in Florida have high RMSE, one is Key Biscayne (25.732°N, 80.163°W) and another is SP_Bayboro (27.762°N, 82.633°W). Both of the two sites contain large portion of urban pixels. The two sites Egbert (44.232°N, 79.781°W) and Toronto (43.790°N, 79.470°W) are only 55 km apart, but the RMSEs have large differences: RMSE at Egbert is 0.09 and that at Toronto is 0.17. The cause of such difference is most likely because Egbert is a rural site and Toronto is an urban site. After applying the bias correction algorithm, all these sites have a reduction in mean bias and RMSE. One exception is the site Grand Forks (47.912°N, 97.325°W) in ND, which has RMSE of 0.17 both before and after the bias correction. The site is found to have large aerosol load from the transport of the western Canada and northwestern US during the time period. Therefore, the large RMSE is caused by uncertainty in aerosol model and is not expected to be significantly reduced by the bias correction algorithm.

Overcorrection, under-correction and/or reduction of correlation are observed at several sites. For example, at NEON_TALL (32.950°N, 87.393°W) in AL, the correlation coefficient decreases from 0.88 to 0.78, the bias decreases from 0.01 to 0, and

RMSE remains 0.06. In the bias correction algorithm, AOD is assumed to hit the background AOD of 0.025 at least once during 30-day period for most of the time steps in order to generate correct curve for bias correction. If this assumption is not satisfied, the algorithm's performance will decrease. If the lowest AOD is higher than the background AOD during the 30-day time period for a pixel for some or all of the time steps, the derived AOD bias curve will be distorted and overcorrection will occur for those time steps. Similarly, slight under-corrections may occur if the lowest AOD during the 30-day period is

lower than 0.025.

Figure 11, analogous to Figure 1, shows the time series comparisons between bias corrected ABI AOD and AERONET AOD for clear days at the same representative AERONET sites used in Figure 1. Almost all of the large biases in Figure 1 are reduced to a magnitude < 0.05 after the bias correction procedure. The exception is in the early morning at the University of

Houston site, where large biases remain. This is probably because the second order polynomial fit of the bias correction does not accurately describe the shape of the AOD biases in this area, which may be the reason why the RMSE of the bias-corrected ABI AOD is still high at the University of Houston site (Table 3, discussed in previous paragraphs).

Figure 12 shows maps of the top 2 qualities of ABI AOD over the Northeast US at 17:42 UTC on October 18, 2018 before

(Figure 12a) and after (Figure 12b) bias correction, illustrating the effects of the bias correction on observed ABI AOD. The black areas in the figures are locations where no AOD was retrieved, primarily caused by cloud coverage. This is a clear day, with no major sources of ambient atmospheric aerosols. However, before the bias correction, Figure 12a shows that the ABI AOD field is noisy, due to the effects of the surface reflectance on the AOD retrievals. For example, over New York City, NY area, uncorrected ABI AOD values are as high as 0.5, while the coincident AERONET AOD measurement at the CCNY

site is only 0.02. After the bias correction, Figure 12b shows that the ABI AOD field is mostly cleared from the surface effects. Some isolated pixels of slightly higher AODs are still observed in the bias corrected ABI AOD map, which are likely originated from cloud contamination, with a few due to incomplete bias correction caused by outliers in fitting the bias correction with a second order polynomial. For comparison, Figure 12 (c) and (d) show MODIS AOD dark target and deep blue retrievals from Aqua for this day, with overpassing time 17:55 UTC. The bias corrected high and medium quality ABI AOD compares well

with MODIS deep blue AOD in both magnitude and data coverage. MODIS dark target AOD has much less data coverage, but ABI AOD also compares well in magnitude in the areas with MODIS dark target AOD data.

Figure 13 shows histograms of original (uncorrected) and bias corrected ABI AOD pixels over the areas within a 27.5 km radius circle around the CCNY AERONET site (Figure 13a) and the Wallops AERONET site (Figure 13b) at 17:42 UTC on

October 18, 2018 (the same observation time as the AOD data shown in Figure 12). At the urban CCNY site, ABI AOD before bias correction ranges from 0 to 0.5, with an average of 0.25, which is much higher than the AERONET AOD value of 0.02. After correction, the ABI AOD distribution narrows down to a very small range with a peak and average at 0.02 - the same value as AERONET. Wallops is a site with mixed pixels of rural, small town and water, and therefore its surface is

darker and more favorable for AOD retrievals. Figure 13b shows that uncorrected ABI AOD at the Wallops site ranges from -0.05 to 0.2, with an average of 0.05, much closer to AERONET AOD (0.03) compared to the matchups at the CCNY site. After the bias correction, the average ABI AOD is 0.03, identical to the AERONET AOD measurement, and the distribution of AOD is narrower than before the bias correction.

As hypothesized, the results discussed thus far suggest that the surface reflectance parameterizations in the ABI AOD algorithm is the main source of the diurnal bias when ABI AOD is close to zero. However, when AOD is higher, such as during periods of high aerosol concentration, the aerosol model in the ABI AOD algorithm becomes a larger source of bias. As an example, a case with a moderate aerosol loading is examined. On August 15-16, 2018, smoke aerosols were transported to the New York City, NY metropolitan area from wildfires burning in the western US and Canada, resulting in AERONET AODs in the range of 0.4-0.7 at the CCNY site. As shown in Figure 14, the bias corrected ABI AOD is very close to the AERONET AOD on August 15 (Figure 14a), but much lower than the AERONET AOD on August 16 (Figure 14b). To investigate the reason for this discrepancy in the bias corrected ABI AOD, the statistics of the ABI AOD retrievals were examined for the 18:12 UTC time step. These statistics are listed in Table 4 for the original ABI AOD pixels within a 27.5 km radius circle of the CCNY AERONET site, which are involved in the average of the matchup with AERONET AOD. AERONET AOD increases from 0.35 on August 15 to 0.55 on August 16, but the uncorrected ABI AOD remains the same on August 16 as on August 15. The reason for this discrepancy is that the aerosol models retrieved within the 27.5 km circle are not the same between the two days. Table 4 indicates that on August 15, the dust model was retrieved primarily (46%), but on August 16, the urban aerosol was predominant. This aerosol event in August 2018 was dominated by smoke, so it is surprising that the ABI AOD algorithm did not select the smoke model a majority of the time on these days. The results for ABI AOD in this case are not unprecedented. The selection of the aerosol model in AOD retrievals over land sometimes does not perform very well in the NOAA VIIRS AOD retrieval either, e.g. over China (Huang et al., 2016; Wang et al., 2020). The ABI retrieval uses only four aerosol models for retrieval over land and the real model may be different from every one of them. Wagner et al. (2018) showed that smoke often carries dust and therefore the aerosol may be a mixture of smoke and dust, which makes the aerosol selection in the AOD retrieval algorithm more challenging, especially because we do not have LUTs for mixtures of aerosols.

Uncertainties in the bias correction algorithm can also be caused by the geometry change within the 30 day period. During 30-day period, the position of the Sun and therefore the solar geometry does change for a given time step. Hence, the surface reflectance relationship and AOD bias are not constant in the time period. The magnitude of AOD bias variation during the time period determines the magnitude of the uncertainty of the algorithm. Besides the change in solar geometry, the surface vegetation color change during seasonal variation may also be a source of uncertainty through its influence on surface reflectance relationships. The choice of 30-day time period to search for lowest AOD in a given pixel was made with extensive research done to minimize the solar zenith angle changes and maximize the chance of finding the lowest AOD (Prados et al., 2007).

## 5.2 Application to NASA DT ABI AOD Data


To demonstrate the effectiveness of the bias correction algorithm and its general applicability, we tested it on an independent geostationary satellite AOD product. The bias correction algorithm is applied to the DT ABI AOD provided by NASA (Gupta et al., 2019). The data used in this study is for the time period of July 2019. Figure 15 shows the scatter plots of DT ABI AOD vs AERONET AOD before and after the bias correction for AERONET sites over CONUS. The original ABI AOD has

a correlation of 0.91, mean bias of 0.07 and RMSE 0.09. After the bias correction, the correlation improves to 0.93, the mean bias reduces to -0.01 and RMSE reduces to 0.05. The original high quality NOAA ABI AOD for the same time period has similar bias and RMSE as the original DT ABI AOD, but has a lower correlation of 0.82 (the scatter plot not shown here). The higher correlation coefficient of DT ABI AOD is probably because DT AOD has lower spatial resolution and DT algorithm selects pixels within 10 km x10 km area by removing the darkest (the darkest 20% over land and 25% over ocean) and the

brightest pixels (the brightest 50% over land and 25% over ocean; Levy et al., 2007; Gupta et al., 2019). The original top 2 qualities NOAA ABI AOD has even lower correlation of 0.79, and it has a mean bias of 0.09 and RMSE of 0.12. After the bias correction, similar to those in Figure 6, both high quality and top 2 qualities NOAA ABI AOD in this time period have correlations of about 0.88, mean biases close to 0 and RMSEs of 0.05.

Figure 16 shows the diurnal variation of the ABI AOD before and after bias correction for three AERONET sites on the days with low AERONET AOD, i.e. GSFC on July 13, 2019, Tucson on July 4, 2019 and CCNY on July 1, 2019. All three sites shows a diurnal variation of the AOD biases. Although Tucson and CCNY only have retrievals certain times of the day, the upward trend in the morning at CCNY and downward trend of in the afternoon at Tucson of dark target ABI AOD are similar to what have been observed in NOAA's ABI AOD product in Figure 1. GSFC has a smaller magnitude of peak at noon than

the other two sites but there is an overall positive bias. The diurnal variation at GSFC is also similar to NOAA ABI data shown in Figure 1 and Figure 2. After the bias correction, biases at all the three sites are reduced. The examples here demonstrate that the biases observed in NOAA's ABI AOD product also exist in other geostationary satellite AOD product because the underlying fundamental question is how well the algorithms can account for surface reflectance contributions to the observed Top of the Atmosphere (TOA) reflectances. The procedures developed for polar-orbiting satellites that worked so well are not

adequate for geostationary satellite geometries. Either the spectral surface reflectance relationships need to be frequently updated in the retrieval algorithms or external empirical bias correction to AOD need to be applied.

Diurnal AOD bias variation pattern was also observed over Asia land surface as well as over ocean when the DT algorithm was applied on Himawari 8 AHI geostationary satellite data (Gupta et al., 2019). The AHI AOD retrieved from DT algorithm

is found to be higher in the morning and lower in the afternoon compared against daily mean. The biases are observed to be

as high as 0.2 and are more serious over ocean for high solar zenith angles. They speculate that the problem may be caused by the errors in radiative transfer code that does not fully account for the curvature of the earth. Although they claim that they did not find any systematic artifact over land, such artifact is expected because it exists in DT ABI AOD over CONUS, as shown in Figure 16 at Tucson and CCNY. Because the bias found in DT AHI AOD is a systematic error, the bias correction algorithm can also potentially be applied on that product, even if it is caused by radiative transfer model.

## 5.3 Impact on Particulate Matter Estimation

NOAA generates AOD products from its polar-orbiting and geostationary satellites for operational use by the National Weather Service as well as the Environmental Protection Agency (EPA) air quality monitoring and forecasting applications. For air quality applications, AOD is often used as a proxy for surface PM2.5 (particulate matter with diameter ≤ 2.5 µm). There are several different ways to scale AOD to surface PM2.5, and the scaling depends on many factors such as relative humidity, boundary layer height, and aerosol composition but the main input is AOD which indicates quantitatively the amount of aerosols present in the atmospheric column. Given that other factors contribute to the regression between AOD and PM2.5, an improved and accurate AOD will influence the accuracy of the estimated surface PM2.5. We tested how the bias correction of ABI AOD improved the PM2.5. Figure 17 shows scatter plots of the correlation between hourly PM2.5 concentration measurements from EPA's ground-based monitor station at Queens College in New York City and GOES-16 ABI AOD before (Figure 17a) and after (Figure 17b) bias correction. The correlation between PM2.5 and ABI AOD improves from 0.58 to 0.68 after the bias correction. These results suggest that applying the bias correction to ABI AOD data will improve its use in air quality monitoring and forecasting applications.

## 6 Analysis of Surface Reflectance and AOD biases

In this section, a further analysis of the behavior of surface reflectance bias and its effect on AOD is performed to demonstrate that it is the source of the AOD bias and the validity of the bias correction algorithm. A radiative transfer simulation is performed using 6SV (Kotchenova et al. 2006; Kotchenova and Vermote 2007) to demonstrate the equivalence of bias correction in AOD and surface reflectance bias reduction.

## 6.1 Surface Reflectance Model Bias Analysis

The surface reflectance relationships used in the operational ABI AOD retrieval algorithm are described in the following equations (ABI AOD ATBD, 2018):

$$\rho_{0.47} = (c_1 + c_2\theta_s) + (c_3 + c_4\theta_s)\rho_{2.25} \qquad\qquad (4)$$

$$\rho_{0.64} = (c_5 + c_6\theta_s) + (c_7 + c_8\theta_s)\rho_{2.25} \qquad\qquad (5)$$

where $\rho_{0.47}$, $\rho_{0.64}$, $\rho_{2.25}$ are surface reflectances at the three bands, $c_1$-$c_8$ are constants depending on NDVI between 0.64 µm and 0.86 µm channel (Equation (1)) as shown in Table 5 (Table 3-12 in the ABI AOD ATBD, 2018), $\theta_s$ is the solar zenith angle.

The coefficients are obtained using a training data set of full disk ABI-AERONET matchup in the time period of April 29, 2017 – January 15, 2018. The reflectances used as training data to generate Equations (4) and (5) were cleared for clouds, screened for low AODs ($< 0.2$) using AERONET AODs, and also used reflectances from ABI pixels within 5 km surrounding the AERONET stations, etc (ABI AOD ATBD, 2018).

Because 0.47 µm band is used in the AOD retrieval algorithm over land, the analysis is focused on 0.47 µm band here. Figure 18 shows the surface reflectance error at 0.47 µm band as a function of scattering angle for three different time periods: (a) April 29,2017 – January 15, 2018, (b) August 6 – December 31, 2018 CONUS, (c) August 6 – December 31, 2018 Full Disk. This is done to test the fidelity of the surface reflectance estimates derived from Equations (4) and (5) when applied to different time periods other than the time period used in the training data as well as when applied to different region of interest. The surface reflectance error is defined as the difference between the surface reflectance at 0.47 µm band estimated using atmospheric corrected 2.25 µm band as input to Equations (4) and (5) and the atmospheric corrected surface reflectance at 0.47 µm band. For the training data set, errors are close to 0 for all the scattering angles, indicating that the fit is good. For the time period August 6-December 31, 2018, which is the time period used in this study, errors are positive for small scattering angles ($<125°$) and negative for larger scattering angles ($> 125°$). There are also some differences between full disk data set and CONUS for the same time period. This figure shows that the behavior of surface reflectance bias is different from what is obtained in the training when the surface reflectance relationship model is applied to a different time period and/or a different region. This is the limitation of the approach of using a universal global surface reflectance relationship model and the reason why a post processing correction of AOD bias is needed unless Equations (4) and (5) is updated regularly. .

## 6.2 Radiative Transfer Simulation Analysis

A radiative transfer simulation study is performed to investigate the AOD retrieval biases due to the surface reflectance errors. A forward calculation is first performed to obtain TOA reflectance with a set of parameters: surface reflectance at 0.47 µm, solar zenith angle, view zenith angle, relative azimuthal angle, AOD, and aerosol model. The surface reflectance is then perturbed with a known bias and AOD is retrieved using the same TOA reflectance. The difference between the retrieved AOD and the input AOD in the forward calculation is the bias due to surface reflectance error. These simulations were performed using the ABI AOD retrieval code and LUT over land (developed based on 6SV radiative transfer model, Kotchenova et al. 2006; Kotchenova and Vermote 2007), in which four aerosol models are used, i.e. generic, urban, smoke,

dust. Standard atmospheric conditions were assumed. The parameters used are listed in Table 6. In the retrieval step, aerosol model is assumed from the four aerosol models. Therefore, there are totally sixteen combinations between the input and the retrieved aerosol models.

The AOD biases obtained in each configuration are grouped by input AOD, surface reflectance, and surface reflectance bias. The mean and standard deviation are calculated and the results are shown in Figure 19. As expected, a negative surface reflectance error introduces a positive AOD error. The corresponding mean AOD bias does not change much with respect to AOD load when AOD is small (less than or equal to 0.5). However, there is a positive increase in the mean bias and a larger standard deviation when AOD is 1.0. This is due to the uncertainty in aerosol model selection. This can be seen in the figure where surface reflectance bias is 0 and the AOD bias is exclusively coming from aerosol model selection error, which tends to give a positive mean AOD bias (about 0.06) and a larger standard deviation (about 0.4).

One can show that the bias correction procedure proposed in this work is valid through this simulation study. In the bias correction algorithm, the AOD bias for a pixel at 0.025 background AOD load is obtained from a 30-day composite procedure, which corresponds to the simulated AOD bias when AOD is 0.025. As shown in Figure 19, the AOD biases at higher AOD are of the similar magnitude as that at 0.025 background AOD if the surface reflectance bias is the same, especially for the negative bias of surface reflectance. For a given pixel, the surface reflectance bias originated from the surface reflectance model is assumed to remain constant during the 30 day period and does not change with AOD load. Therefore, AOD biases at higher AOD load can be estimated by the AOD bias obtained at background AOD of 0.025.

## 7 Summary and Conclusions

In our validation work of GOES-16 ABI AOD, we noticed a substantial diurnal bias in AOD that needed to be fixed for our operational users. Analysis shows that the bias is caused by errors in the land surface reflectance relationship between the spectral bands used in the ABI AOD retrieval algorithm. To remove the biases, an empirical algorithm is developed that utilizes the lowest AOD in a recent 30-day period in conjunction with the background AOD to derive a smooth bias curve at each ABI AOD pixel. The ABI AODs are then corrected by subtracting the derived bias curves at each time step.

The bias correction algorithm is validated for five months of GOES-16 ABI AOD data through comparisons against coincident AERONET AODs. The results demonstrate that the bias correction algorithm works successfully: for the top 2 qualities of ABI AODs, the correlation with AERONET AOD, average bias, and RMSE all improve. As a result of the bias correction, top 2 qualities ABI AOD performs as well as uncorrected high-quality ABI AOD. Therefore, bias corrected top 2 qualities ABI AOD data are recommended for use in research and operations. The bias corrected AODs cover twice the area as high-quality ABI AOD data alone with the same accuracy.

The ABI AOD bias correction process is most effective when AOD is low because under those conditions, the surface reflectance relationship is the main source of uncertainty in the ABI AOD retrieval. When AOD is higher, the uncertainty from the aerosol model selection in the ABI AOD retrieval algorithm becomes as large as or larger than that from the surface

reflectance relationship, and therefore the bias correction for high AOD conditions is not as effective as that for low AOD conditions.

The surface reflectance relationships in the ABI AOD retrieval algorithm will be improved when more GOES-16 data are accumulated and analyzed. However, these relationships are based on AERONET sites and they are statistical models.

Therefore, individual AOD pixels will always suffer to some degree from deviation in the statistical relationship and some bias will always exist, although it may be reduced by a more accurate surface reflectance relationship. Hence, future versions of the GOES ABI AOD product may still benefit by applying the bias correction algorithm, unless the AOD retrieval algorithm uses pixel level surface reflectance relationships that are routinely updated. Such an exercise in an operational setting is prohibitive


The bias correction algorithm has a general applicability. It can also be applied to other geostationary AOD products, which may also suffer the bias described in this research; especially, if the AOD algorithms are similar relying on deriving surface reflectance from relationships between blue band and SWIR band. We tested and demonstrated that the performance of

NASA's dark target ABI AOD is improved by applying the bias correction algorithm. The existence of bias in NASA's dark target algorithm indicates that the bias issue is a more general problem rather than only existing in NOAA's ABI AOD product. Therefore, other geostationary AOD products can benefit by applying the bias correction technique introduced in this research.

**Data Availability**

GOES-16 ABI AOD can be obtained at NOAA CLASS (https://www.avl.class.noaa.gov/ ; accessed on 5/29/2020).

AERONET AOD can be obtained at https://aeronet.gsfc.nasa.gov/ (accessed on 5/29/2020). The data produced from the bias correction algorithm can be requested by contacting Hai Zhang (hai.zhang@noaa.gov). The bias corrected ABI AOD product will be implemented and available in near-real-time on NOAA's data server.

## Author Contributions

HZ worked on the developing and analyzing activities described and led the manuscript writing. MZ worked on surface reflectance relationship analysis. SK and IL are co-leads of the aerosol algorithm development and guided the work. SK, IL and MZ reviewed the algorithm science and analysis, and contributed to the paper revisions. MZ and IL provided the AOD retrieval code that is used in the atmospheric correction for the surface reflectance analysis.

## Competing interests

The authors declare that they have no conflict of interest.

## Acknowledgements

The authors thank the AERONET principal investigators and site managers for providing the data used in this work and Amy Huff (IM Systems Group) for providing internal review. The authors thank NASA DT (Rob Levy and Pawan Gupta) for 625 providing their DT GOES-16 ABI AOD retrievals. The contents of this paper are solely the opinions of the authors and do not constitute a statement of policy, decision, or position on behalf of NOAA or the U. S. Government.

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

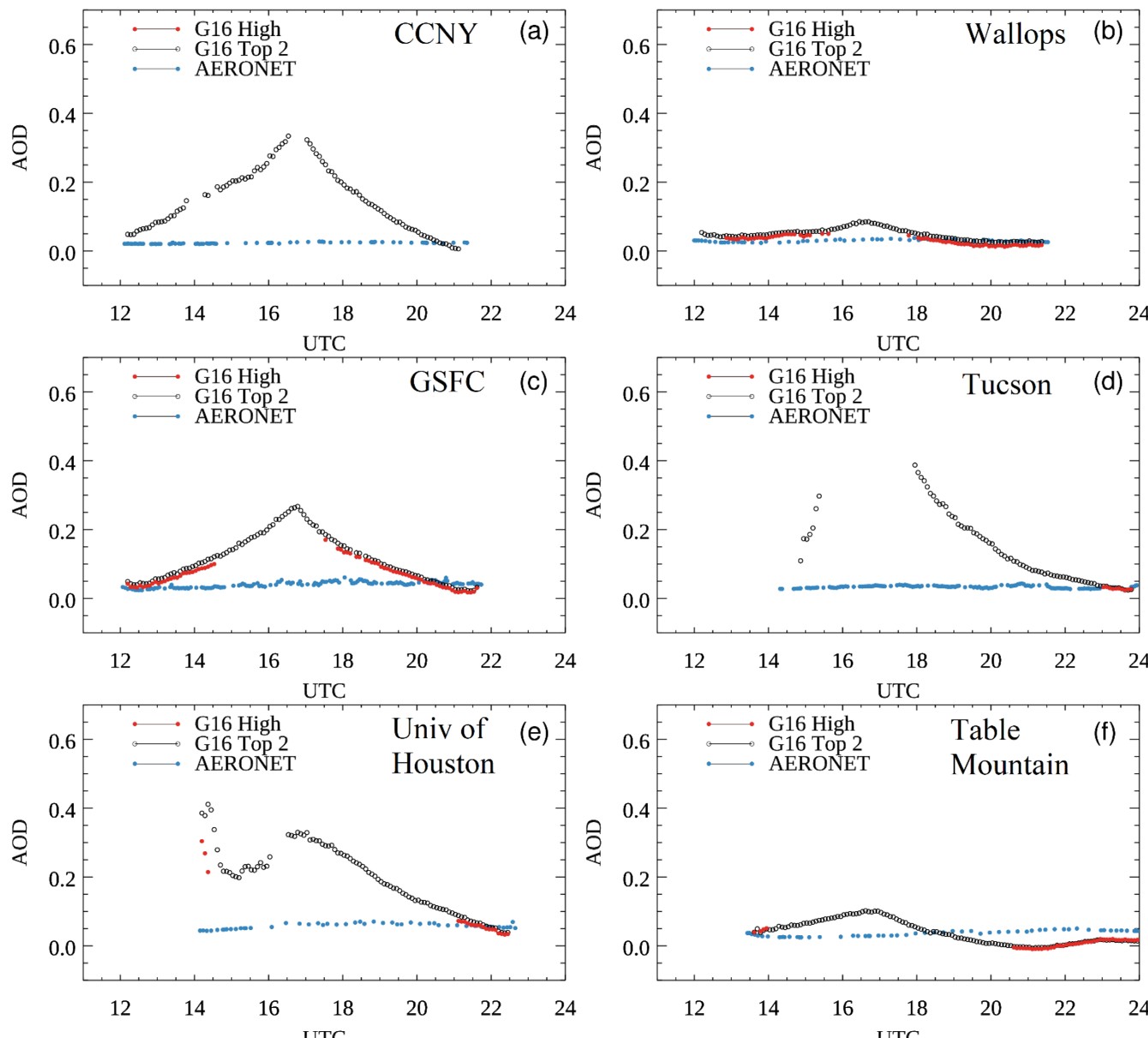

**Figure 1:** Time series of GOES-16 ABI AOD and AERONET AOD at 6 representative AERONET sites: (a) CCNY on October 18, 2018, (b) Wallops on October 18, 2018, (c) GSFC on October 12, 2018, (d) Tucson on October 25, 2018, (e) University of Houston on December 22, 2018, and (f) Table Mountain on September 12, 2018, showing the diurnal variations in the ABI AOD bias. Details about the AERONET sites are listed in Table 2. Clear days are selected such that AERONET AOD are ≤ 0.05 throughout the entire day. "G16 High" represents GOES-16 high quality AOD and "G16 Top 2" represents GOES-16 high quality and medium quality AOD.

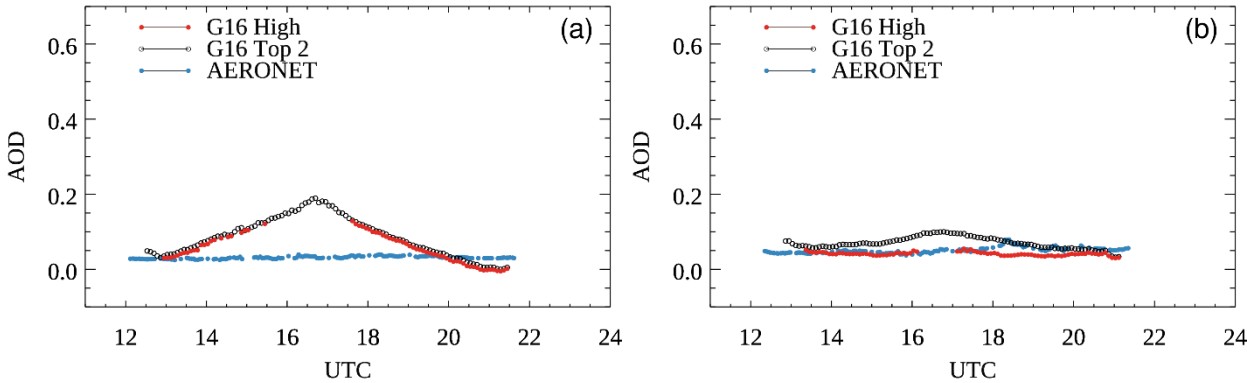

**Figure 2. The diurnal pattern of biases in GOES-16 ABI AOD at GSFC on two additional clear days: (a) October 18, 2018 and (b) October 30, 2018, showing the difference in the magnitude of the bias.**


**Figure 3. Scatter plots of surface reflectance on 0.47 μm band and 2.2 μm band for three days, i.e. October 12$^{th}$, October 18$^{th}$, and October 30$^{th}$ 2018, at GSFC at (a) 17:02 UTC and (b) 20:02 UTC, and histograms of NDVI for the three days at (c) 17:02 UTC and (d) 20:02 UTC. The lines on the scatter plots are the surface reflectance relationship between 0.47 μm band and 2.2 μm band used in the ABI AOD retrieval algorithm.**

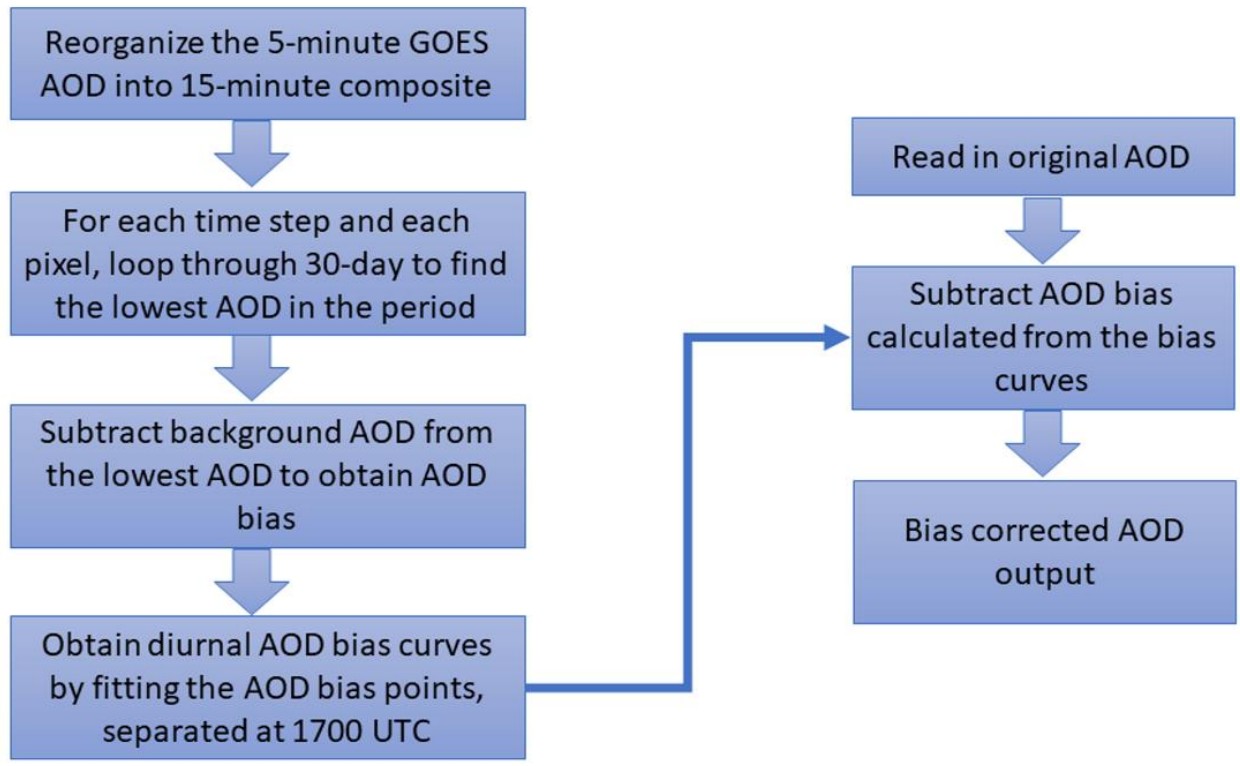


**Figure 4. Flow chart of the ABI AOD bias correction algorithm.**

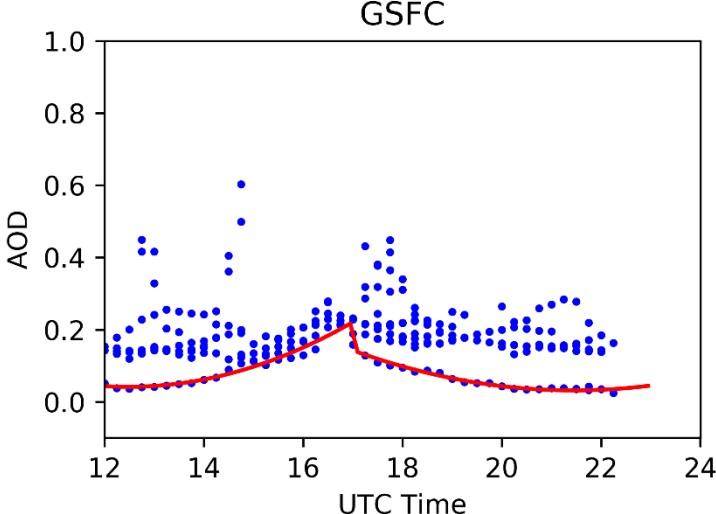


**Figure 5. AOD at a pixel close to GSFC over time period of September 12 – October 11, 2018 (blue dots) vs UTC time and the lower bound of the AOD (red curve).**

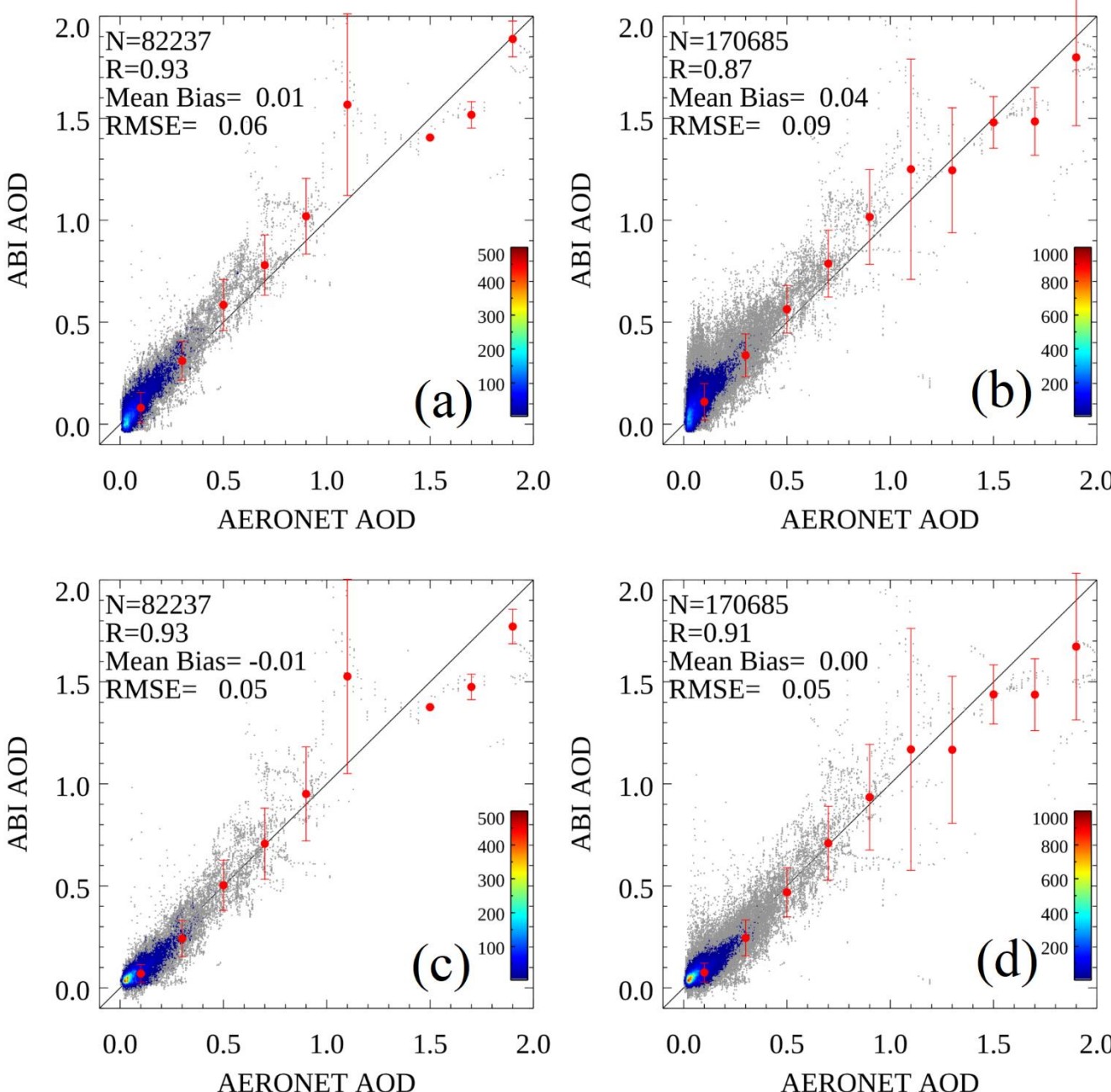

**Figure 6.** Scatter plots of GOES-16 ABI AOD vs AERONET AOD for August 6, 2018 to December 31, 2018 across the CONUS domain: (a) high quality ABI AOD before bias correction, (b) top 2 qualities ABI AOD before bias correction, (c) high quality ABI AOD after bias correction, and (d) top 2 qualities ABI AOD after bias correction. The red circles and vertical bars are the mean ABI AOD and the standard deviation of errors of data points falling in the bins with size of 0.2. In the plots, N is the number of matchups, R is the correlation coefficient, and RMSE is the root mean square error.

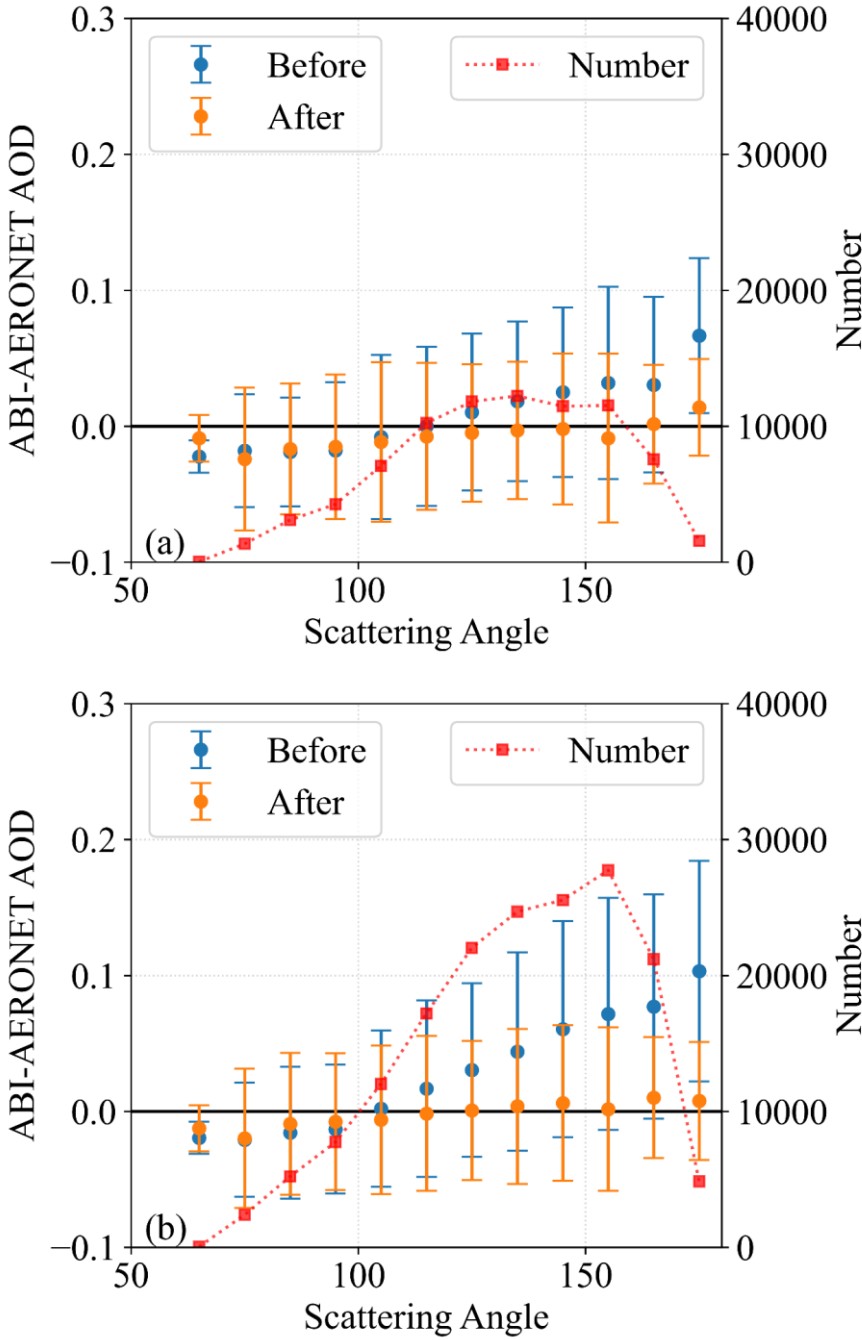


**Figure 7. Comparisons of ABI AOD error vs scattering angle between before and after bias correction for (a) high quality and (b) high and medium quality.**

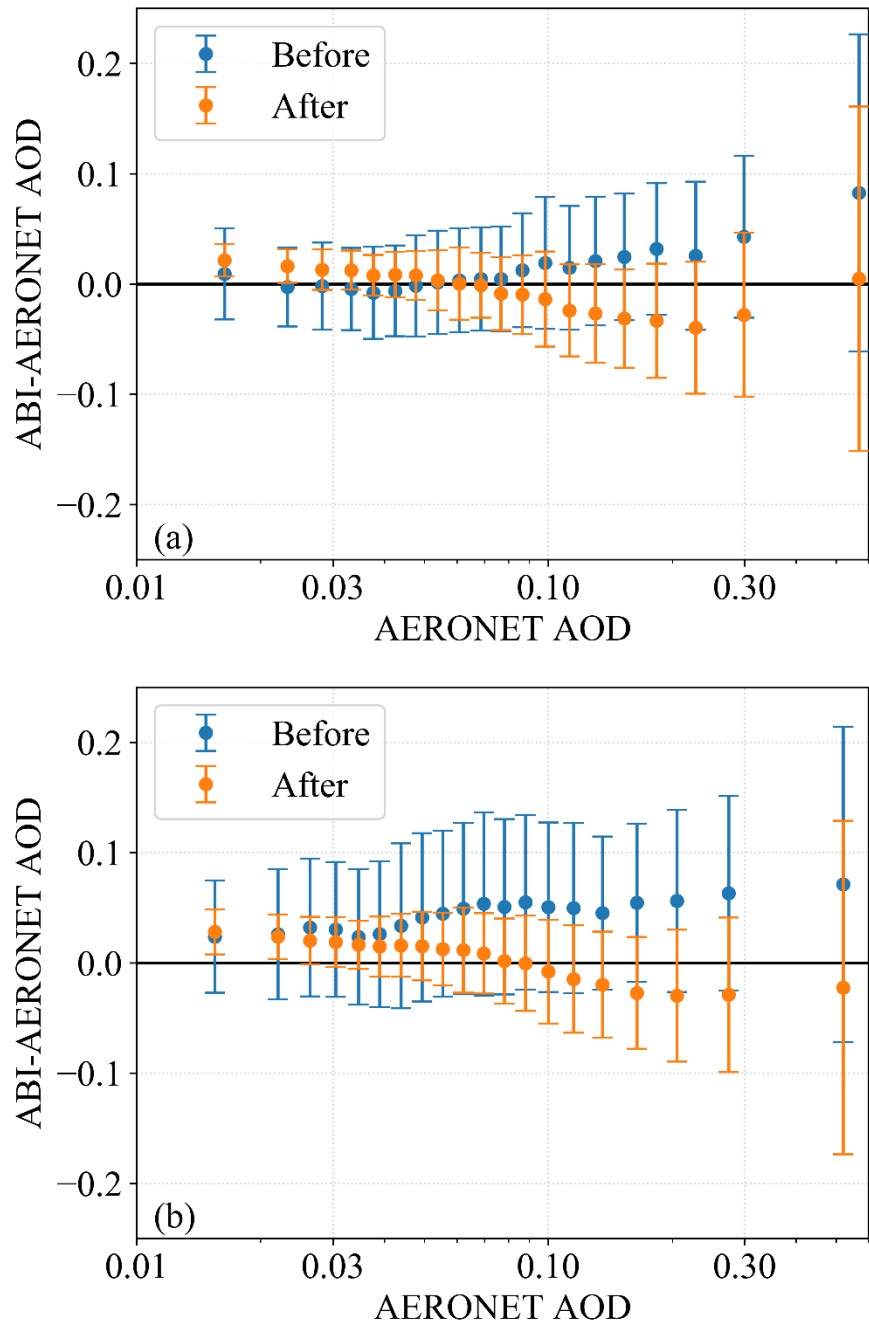

**Figure 8. Comparisons of ABI AOD error vs AERONET AOD between before and after bias correction for (a) high quality and (b) high and medium quality. Each bin contains equal number of matchup data.**


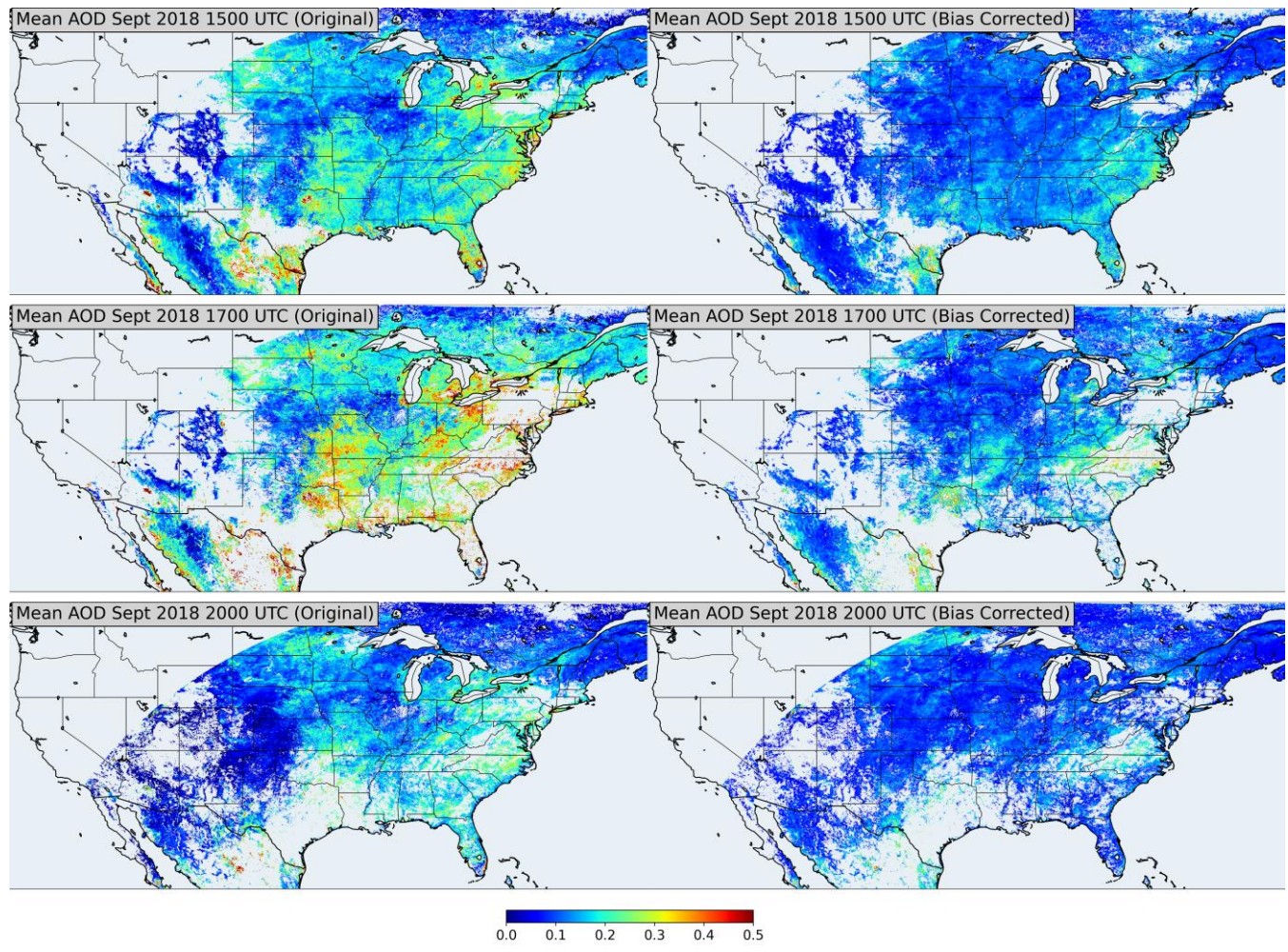

**Figure 9.** Monthly mean AOD (top 2 qualities) for September 2018 at three time steps, i.e. 1500 UTC, 1700 UTC and 2000 UTC, for the original ABI AOD (left column) and bias corrected AOD (right column).

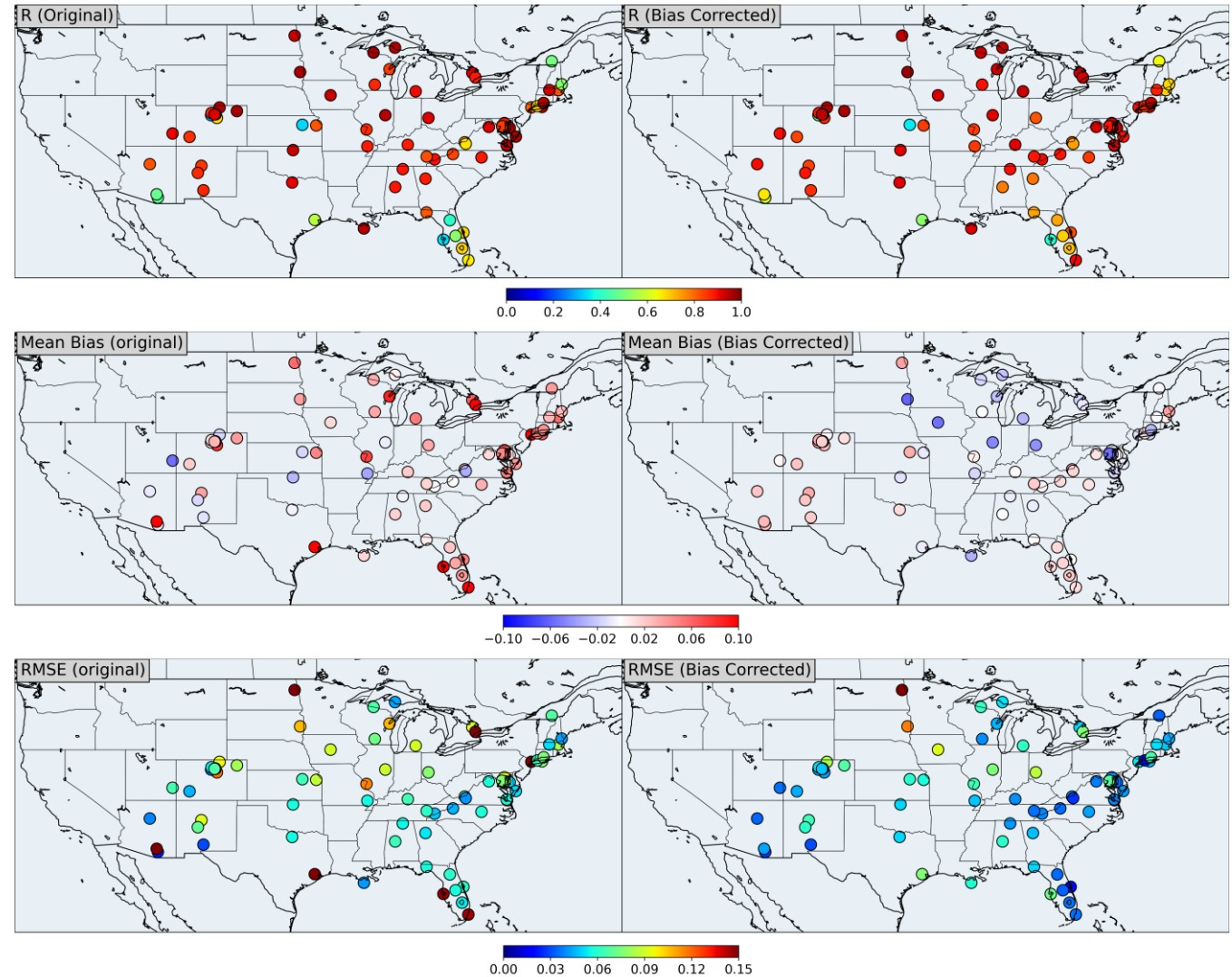

**Figure 10.** **Maps of correlation coefficients, mean biases and RMSEs at AERONET sites with more than 400 matchups for the time period August 6-December 31, 2018 for the original ABI AOD (top 2 qualities) vs AERONET AOD (left column) and for the bias corrected ABI AOD (top 2 qualities) vs AERONET AOD (right column).**


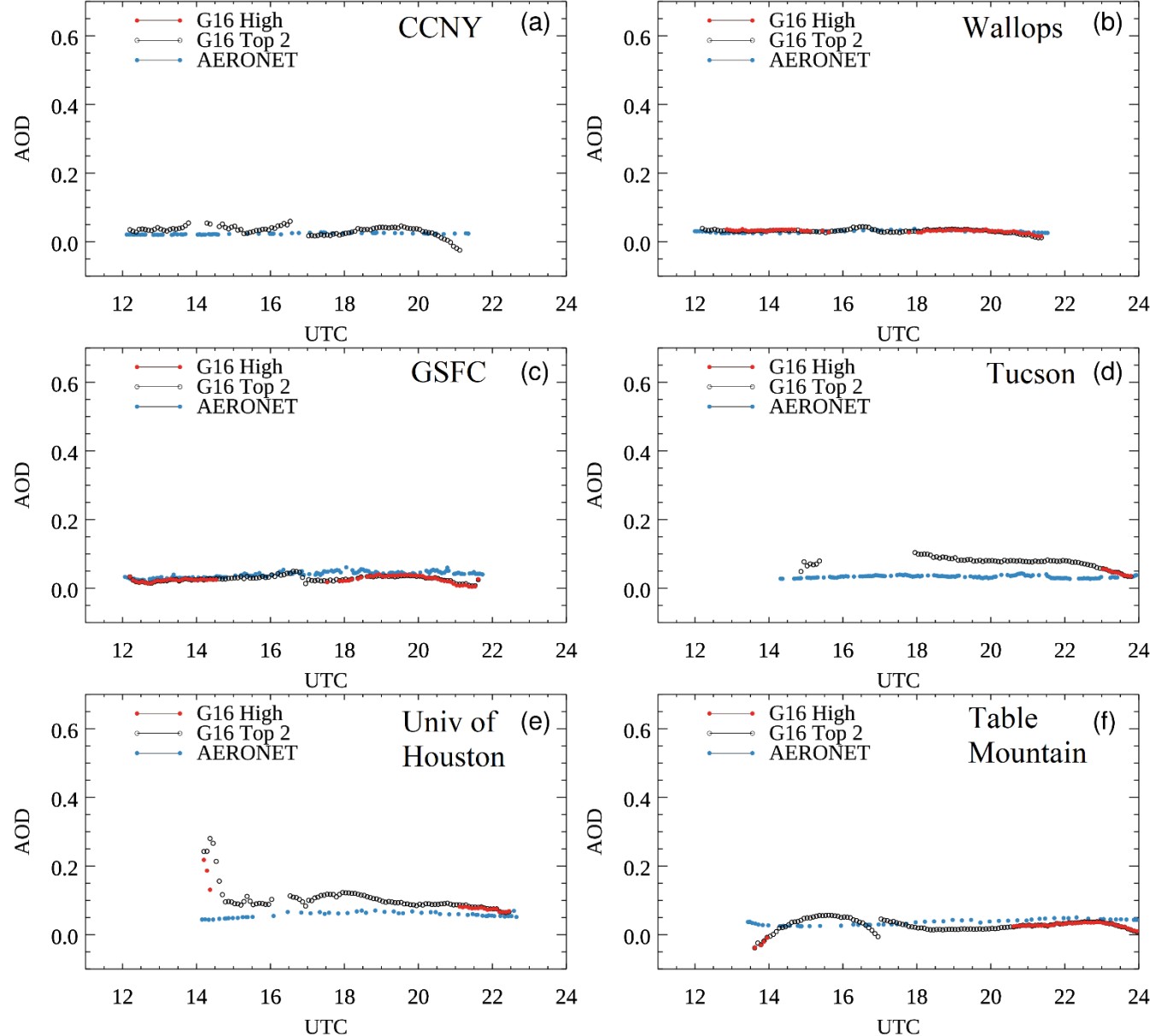

**Figure 11. Same as in Figure 1, but after correcting the GOES-16 ABI AOD for the diurnal bias.**

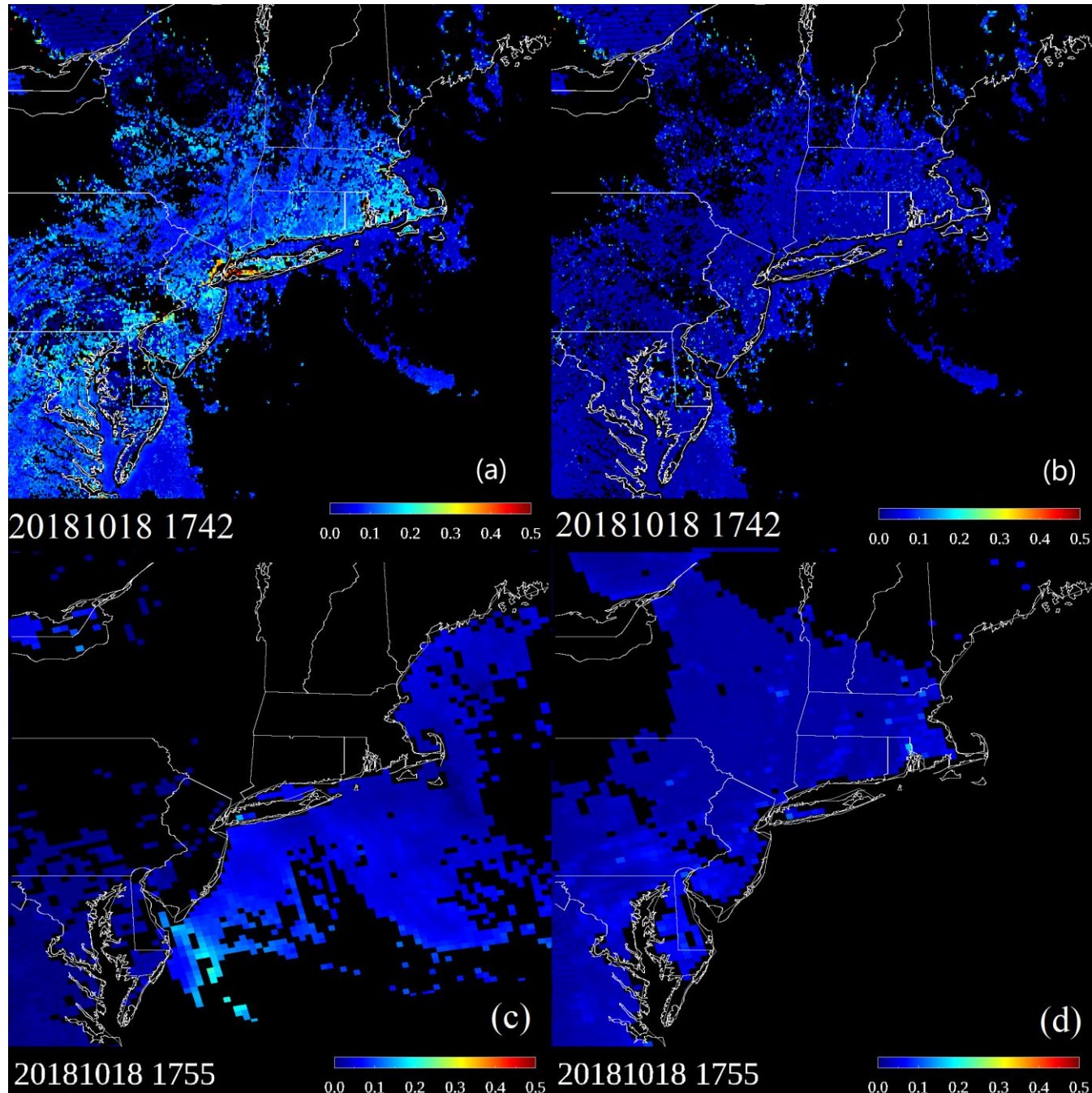

**Figure 12. Maps of GOES-16 ABI AOD, top 2 qualities (high and medium), over the Northeast US at 1742 UTC on October 18, 2018: (a) before bias correction and (b) after bias correction, and high quality MODIS Aqua AOD (c) dark target product and (d) deep blue product.**



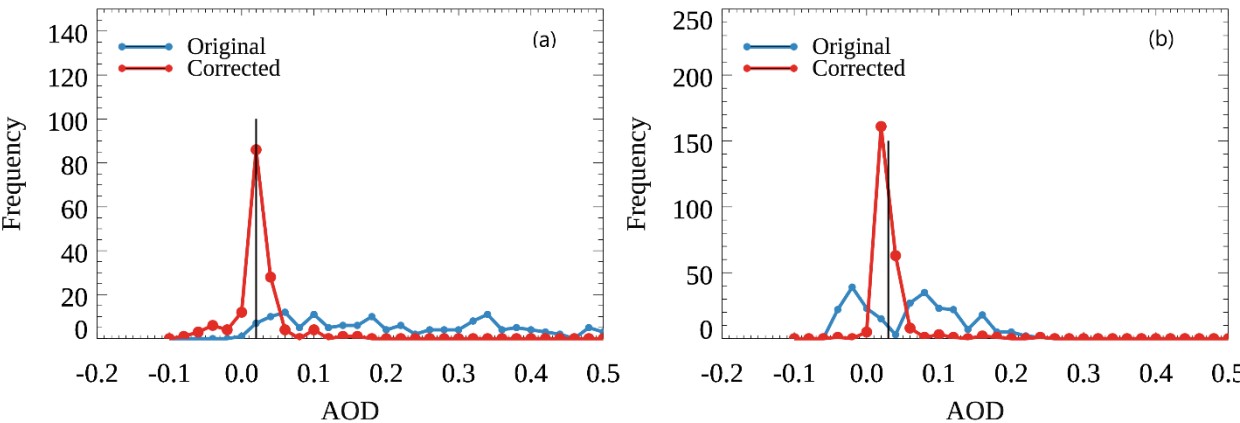

**Figure 13.** Histograms of original (uncorrected) and bias corrected GOES-16 ABI AOD at the (a) CCNY and (b) Wallops AERONET sites, at 17:42 UTC on October 18, 2018. The black vertical lines in the figures represent AERONET AODs.

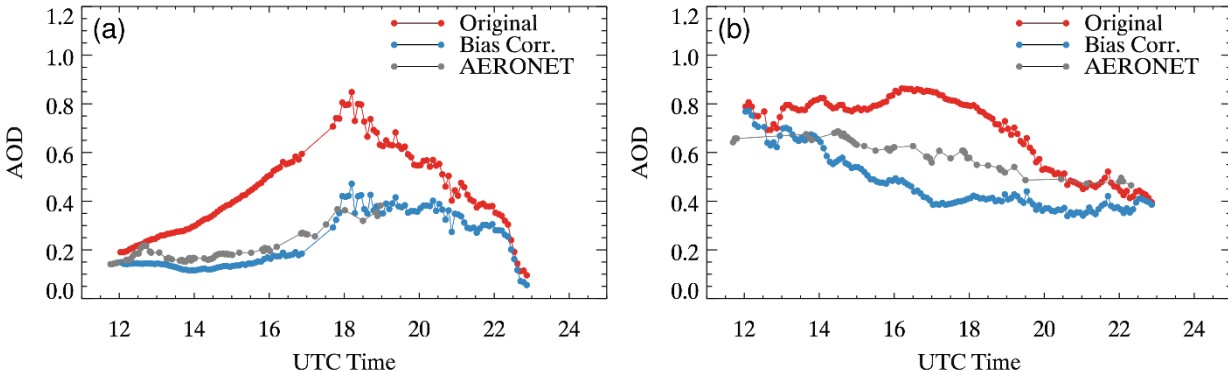

**Figure 14. Time series of original (uncorrected) GOES-16 ABI AOD, bias corrected ABI AOD, and AERONET AOD at the CCNY AERONET site on (a) August 15, 2018 and (b) August 16, 2018, showing the difference in bias corrected ABI AOD relative to AERONET AOD on two consecutive days with moderate aerosol loading.**



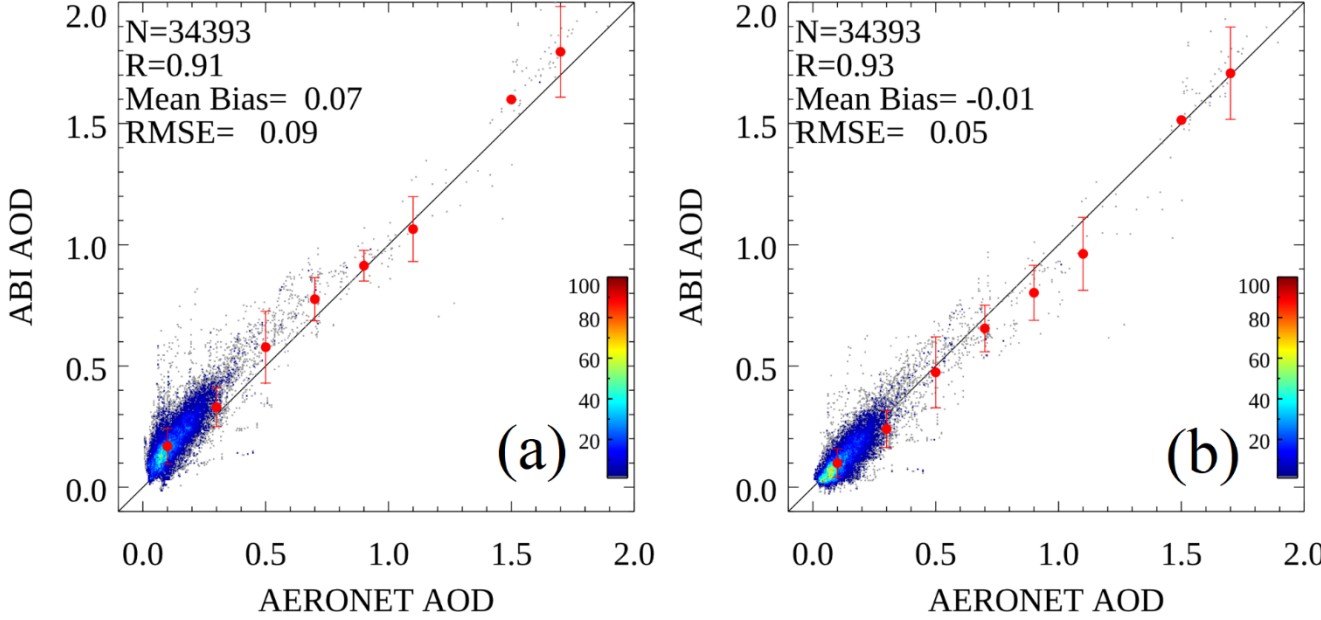

**Figure 15. Scatter plot of NASA's dark target ABI AOD vs AERONET AOD for July 2019 over CONUS: (a) original; (b) bias corrected.**

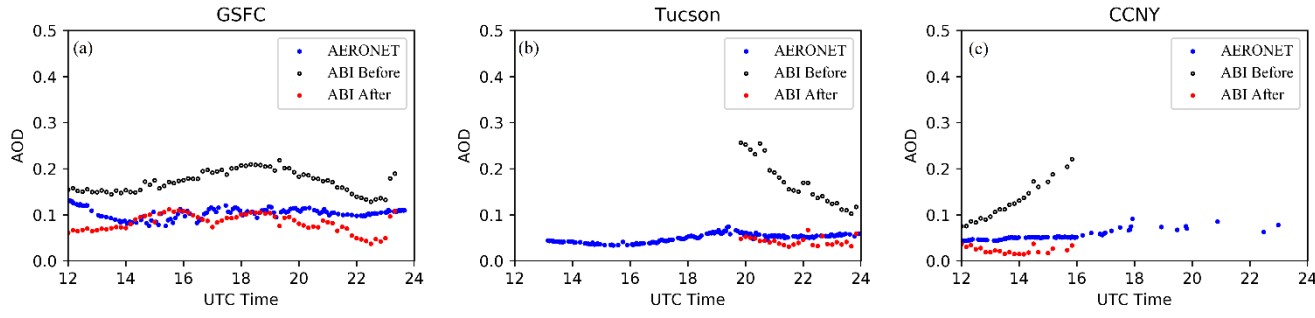

**Figure 16. Diurnal time series over three selected AERONET sites and days with low AERONET AOD for NASA ABI AOD (a) GSFC July 13, 2019; (b) Tucson July 4, 2019; (c) CCNY July 1, 2019.**

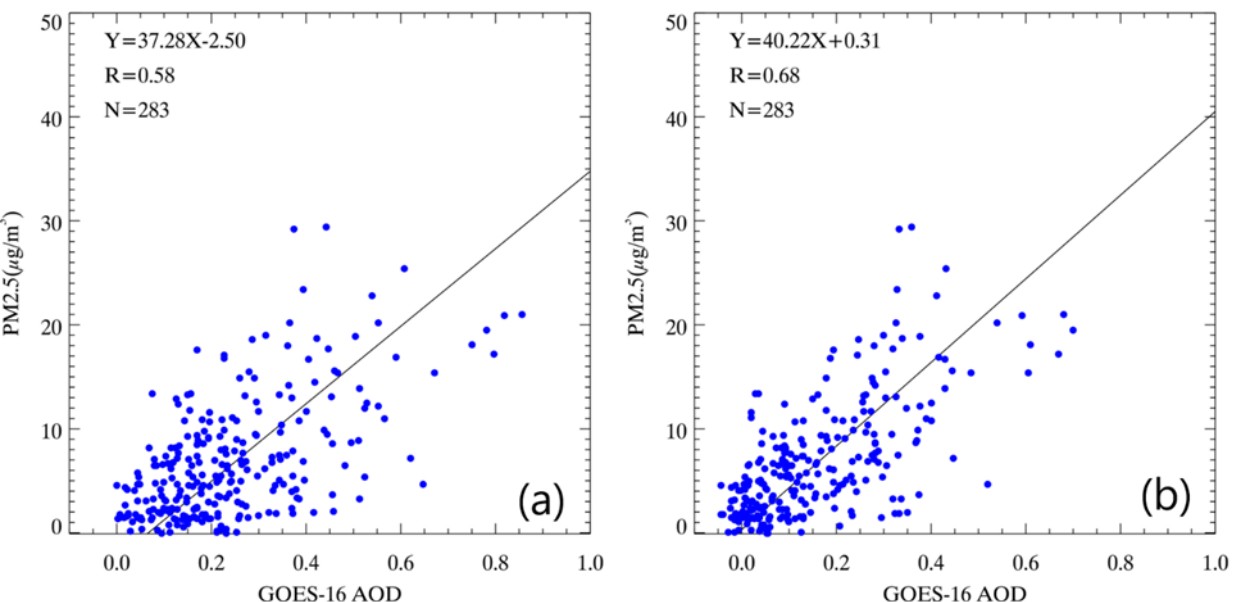

**Figure 17.  Scatter plots of hourly PM2.5 vs GOES-16 ABI AOD at an EPA station at Queens College in New York City during August 6 – December 31, 2018: (a) GOES-16 ABI AOD before bias correction; (b) GOES-16 ABI AOD after bias correction.**


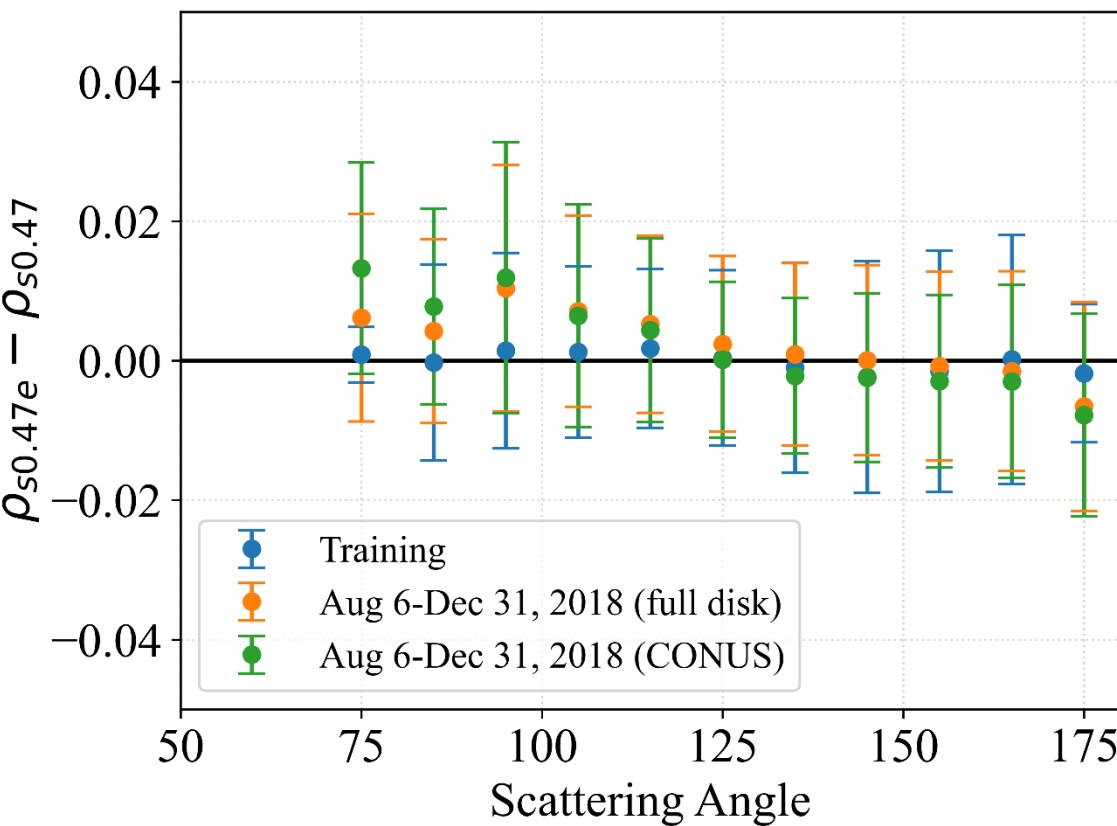

**Figure 18.** Surface reflectance error at 0.47 μm band vs scattering angle for training data set, Aug 6-Dec 31, 2018 full disk data, and Aug 6-Dec 31, 2018 CONUS data.

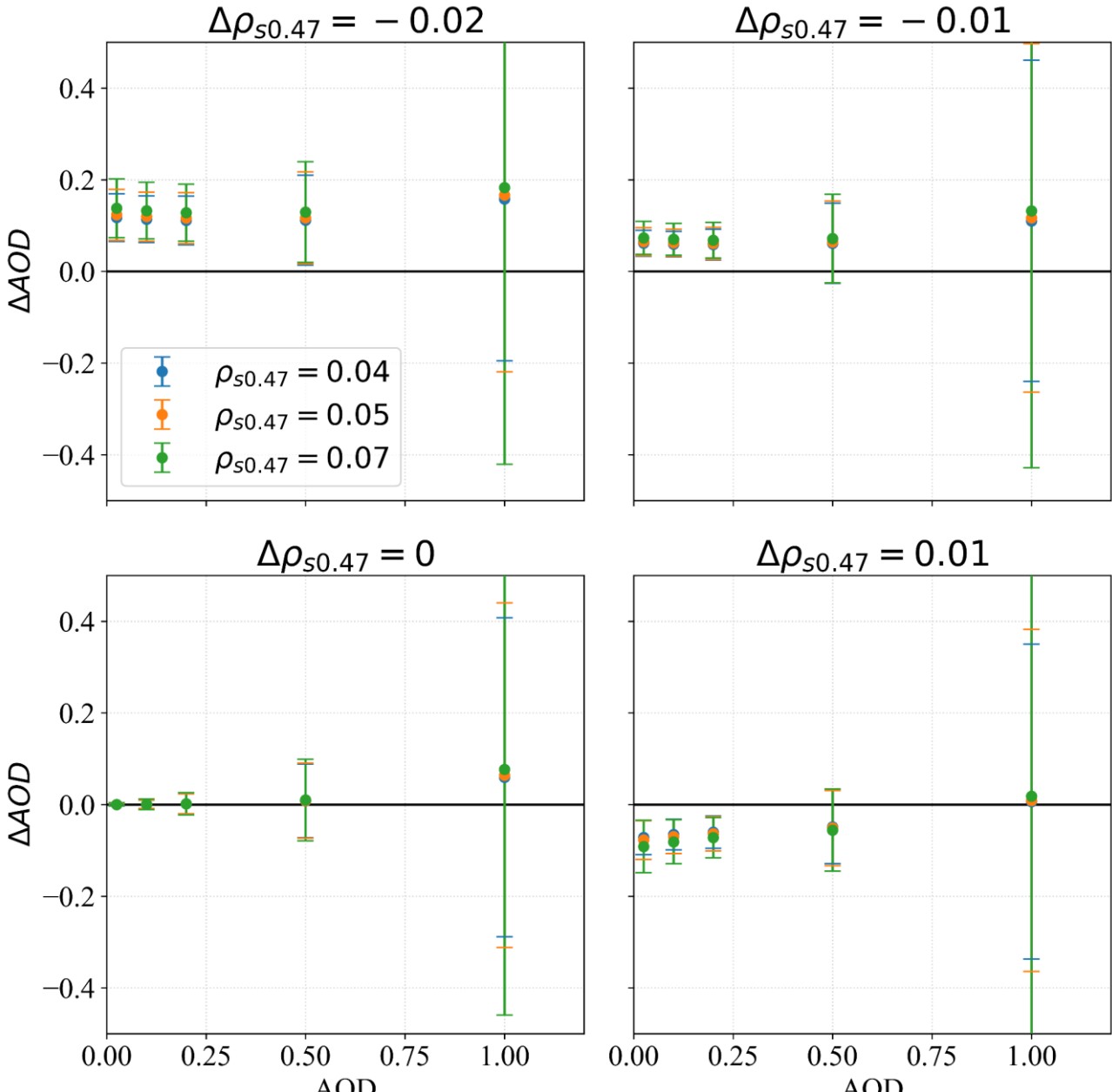


**Figure 19 AOD retrieval uncertainty due to the uncertainty of surface reflectance at 0.47 μm band vs AOD load.**

| Quality Level | Condition |
|---|---|
| No retrieval | Invalid input data, Cloud, Snow/ice, Bright land surface, Sun glint over ocean |
| Low | External and internal cloud tests contradict, Low satellite (satellite zenith angle > 60°), Low sun (solar zenith angle > 80°), AOD out of range, Coastal, Shallow inland water, High residual, High inhomogeneity |
| Medium | Cloud/Snow adjacency, Shallow ocean, Probably clear, Medium inhomogeneity, Medium residual |
| High | Remaining |

**Table 1. Conditions for quality levels of ABI AOD pixels.**

| Site Name | Location | Coordinates | Type |
|---|---|---|---|
| City College of New York (CCNY) | New York City, NY, USA | 40.821°N, 73.949°W | Urban |
| Wallops | Wallops, MD, USA | 37.933°N, 75.472°W | Mixed rural, small town, and water. |
| Goddard Space Flight Center (GSFC) | Greenbelt, MD, USA | 38.992°N, 76.839°W | Suburban |
| Tucson | Tucson, AZ, USA | 32.233°N, 110.953°W | Urban |
| University of Houston | Houston, TX, USA | 29.717°N, 95.341°W | Urban |
| Table Mountain | Longmont, CO, USA | 40.125°N, 105.237°W | Rural |

**Table 2. Details about the representative AERONET sites used as examples to illustrate the range of the observed diurnal bias in GOES-16 ABI AOD.**


| Site | N | R | | Slope | | Intercept | | Bias | | RMSE | |
|---|---|---|---|---|---|---|---|---|---|---|---|
| | | Before | After | Before | After | Before | After | Before | After | Before | After |
| City College of New York (CCNY) | 2810 | 0.81 | 0.89 | 1.40 | 0.78 | 0.07 | 0.01 | 0.12 | -0.01 | 0.15 | 0.05 |
| Wallops | 4267 | 0.95 | 0.89 | 1.16 | 0.74 | 0.02 | 0.02 | 0.04 | -0.01 | 0.05 | 0.04 |
| Goddard Space Flight Center (GSFC) | 3972 | 0.86 | 0.90 | 1.33 | 0.91 | 0.02 | 0.00 | 0.06 | -0.01 | 0.09 | 0.04 |
| Tucson | 4507 | 0.47 | 0.66 | 3.64 | 1.22 | -0.01 | 0.02 | 0.11 | 0.03 | 0.16 | 0.04 |
| University of Houston | 2197 | 0.57 | 0.52 | 1.95 | 1.10 | 0.05 | -0.02 | 0.15 | -0.01 | 0.19 | 0.08 |
| Table Mountain | 3695 | 0.92 | 0.94 | 1.19 | 1.06 | 0.01 | 0.01 | 0.03 | 0.02 | 0.07 | 0.05 |

**Table 3. Validation statistics for comparisons between GOES-16 ABI AOD (top 2 qualities) and AERONET AOD at the 6 representative AERONET sites listed in Table 2 for August 6, 2018 to December 31, 2018 across the CONUS domain, both before and after bias correction. N is the number of matchups, R is the correlation coefficient, and RMSE is the root mean square error.**

| | Average GOES-16 AOD | Total number of pixels | AERONET AOD | Dust | | Generic | | Urban | | Heavy smoke | |
|---|---|---|---|---|---|---|---|---|---|---|---|
| | | | | N (%) | AOD | N (%) | AOD | N (%) | AOD | N (%) | AOD |
| 20180815 | 0.82 | 41 | 0.35 | 19 (46%) | 0.87 | 3 (7%) | 0.84 | 10 (24%) | 0.51 | 9 (21%) | 1.05 |
| 20180816 | 0.80 | 246 | 0.55 | 50(20%) | 1.04 | 28 (11%) | 0.74 | 101 (41%) | 0.60 | 67 (27%) | 0.94 |

**Table 4. Statistics of original (uncorrected) ABI AOD and AERONET AOD retrievals at the CCNY AERONET site on August 15**
**and 16, 2018 for the 4 aerosol models used in the ABI AOD algorithm.**

| NDVI ranges | $c_1$ | $c_2$ | $c_3$ | $c_4$ | $c_5$ | $c_6$ | $c_7$ | $c_8$ |
|---|---|---|---|---|---|---|---|---|
| NDVI $\geq$ 0.55 | 1.436330E-02 | 2.060893E-04 | 1.749239E-01 | -2.859502E-03 | 1.374160E-02 | -5.128175E-05 | 2.761044E-01 | 1.034823E-03 |
| 0.3 $\leq$ NDVI < 0.55 | 4.163894E-02 | -2.147513E-04 | 1.598440E-01 | 7.401292E-04 | 2.990101E-02 | -1.873911E-04 | 4.602174E-01 | 9.658934E-04 |
| 0.2 $\leq$ NDVI < 0.3 | 5.154307E-02 | 5.679386E-05 | 2.048702E-01 | -7.064656E-04 | 5.179930E-02 | -1.043257E-04 | 4.937035E-01 | 4.310074E-04 |
| NDVI < 0.2 | -4.990575E-02 | 2.138207E-03 | 8.498076E-01 | -1.179596E-02 | -3.397737E-02 | 1.640336E-03 | 1.087497E+00 | -9.538776E-03 |

**Table 5. Surface reflectance relationship coefficients used in Equations (4) and (5). (Table 3-12 of ABI AOD ATBD 2018)**



| Parameters | Values |
|---|---|
| Solar zenith angle | 30°-60° with 10° interval |
| View zenith angle | 30°-60° with 10° interval |
| Relative azimuthal angle | 0°-180° with 20° interval |
| Aerosol model | Generic, urban, smoke, dust |
| Surface reflectance at 0.47 µm | 0.04, 0.05, 0.07 |
| AOD | 0.025, 0.1, 0.2, 0.5, 1.0 |
| Surface reflectance bias at 0.47 µm | -0.02, -0.01, 0, 0.01 |

**Table 6. Parameters used in AOD uncertainty simulation due to the surface reflectance uncertainty.**