# Peer review of "Improving GOES Advanced Baseline Imager (ABI) Aerosol Optical Depth (AOD) Retrievals using an Empirical Bias Correction Algorithm"

_Atmospheric Measurement Techniques, 2020_

## Referee Comment (RC1) · Anonymous Referee #1 · 15 Apr 2020

The goal of this paper is to correct for AOD retrieval biases in GOES ABI AOD product using an empirical approach.

The surface reflectance in the current GOES AOD algorithm is estimated based the relationship between 0.47 and 2.2 um and 0.64 and 2.2 um since most aerosols are 'transparent' in the 2.2 um. This is based on the Kaufman et al (1997, IEEE) paper that many MODIS algorithms use to estimate surface reflectance. In this paper, the authors look for 'clear days' (based on AERONET AOD values less than 0.05) to assess the GOES AOD (for both high and medium quality retrievals) for a few selected sites. They report that the GOES AOD is biased since the GOES AOD is much larger than the

AERONET AOD for these clear days. The authors note that the biases appear to be centered around 1700 UTC and it is due to surface reflectance parametrizations at various sun-satellite viewing geometries. The authors then attempt to correct this bias based on the premise that it is the surface reflectance that is the issue in the GOES algorithm. Then they use a 30-day composite of GOES AOD to estimate the minimum AOD and subtract that with the background AOD (a fixed value of .025) to correct for the bias. They use two polynomial fitted relationships to estimate biases. They then correct the AOD using these relationships and then validate the results with AERONET AOD and show improvement in these biases.

First I need to note that the paper needs to go through some editorial clean up since several sentences are awkward; key references (Kondragunta et al 2020) are missing; and some references are really old.

I find several problems with the paper and most importantly it is the use of AOD to make these corrections rather than working with the reflectances. The algorithm retrieves AOD based on apriori assumptions of aerosol model, surface parametrizations based on NDVI , cloud clearing approaches and a host of thresholds for cloud cover, inhomogeneity, etc (ATBD, 2018). Now this paper indicates that the surface parametrizations are a problem and then to remove the biases the authors use the retrieved AOD to make bias adjustments. The original algorithm uses reflectance ratios to arrive at surface values and now this paper goes back to the older GASP approach to obtain the 30-day composite minimum (not reflectance) AOD values. Looking at this from an algorithm perspective it is not the correct solution for an operational algorithm to go through retrieval using one set of processes, retrieve AOD's and then use the retrieved AOD values to make corrections for parameters that are part of the original retrieval process (in this case surface reflectance). The authors need to think about having the correct algorithm as part of the retrieval process rather than adjusting it after the retrieval is done.

The other issue is the relaxation of quality flags to allow more data. There were strong

reasons for picking all the metrics for high and medium quality flags in the first place (ATBD, 2018) whether it is cloud/snow cover or inhomogeneity. Line 90 to 95 provides the various reasons for selecting the pixels for the retrievals and this paper now allows all the medium quality flags in the process but does not address cloud contamination issues. The paper needs to be more convincing that it is indeed surface issues and not cloud cover that causes these problems. The results need to be discussed in terms of scattering angles (see She et al, Remote Sensing, 2019). This will allow more quantitative analysis rather than statements like those in 160-161. Also for Figure 1 and Figure 2 what were the histograms of actual reflectance's from the GOES channels for the various peaks. This can help explain Figure 2 better.

The paper uses two sets of parametrizations for adjusting the biases and then in line 255 back tracks the approach by stating that this could have large uncertainties.

Figure 8 and Figure 9 appears as a complete afterthought since the aerosol model discussion is not complete or convincing. I have no idea why the PM2.5 discussions (Figure 9) is relevant for this paper.

The AERONET data used is from 2018 and the authors need to be using Level 2 not 1.5. This data should be available.

Other issues. Define accuracy and precision and be quantitative rather than merely stating that one product is better than the other. Line 50, Deemed to have quality sufficient is rather vague. Line 73: The word transparent to most aerosols is rather vague. Describe why this is possible briefly based on aerosol size and extinction Line 74. Again, poor phrasing. It is not linear reflectance BETWEEN channels if it is three channels. Be specific. Line 75-80 is awkward phrasing. The algorithm does not make retrievals? Describe the algorithm clearly but briefly. Line 81-84 is not clear at all. While I understand how the algorithm works this type of writing will not help all readers understand the algorithm and methods used in this paper. Line 89: Usually very small? What does that mean? Need some numbers. Lines 89-94 needs to be clearer with

brief discussion rather than listing the problems. Line 98: If the surface reflectance issues are so different between 0.41 and 0.47 micron then the authors need to show or discuss this for certain land types. Otherwise these statements are vague. 115-120 discussion is not "technical" enough. What does air mass movements mean? You need to then state what wind speeds at what height provide the 27.5 km radius. Line 140+: How about retrieval biases due to sun-satellite viewing geometry in radiative transfer code? Line 147: We need to see these relationships between two channels for the solar geometries.

I find the two reasons in 152-155 to be problematic. Why should the test position issue matter if these relationships are established for certain solar viewing geometries/NDVI? Plus there are reasons why the quality flags were established for high, low, medium in the first place (cloud cover, snow cover etc). Of course one would use the best quality flags for establishing surface reflectance relationships because of contamination issues. Now if you are using medium quality flags to get more data into the analysis then of course your surface reflectance relationships are going to be different.

Since this paper is about surface reflectance issues the authors need to show these relationships that currently exist for various angles/NDVI first to make their case stronger.

The authors should also show the reflectance values on these plots so we can interpret the results better.

---

## Referee Comment (RC2) · Anonymous Referee #2 · 16 Apr 2020

This paper evaluates the AOD retrieval from geostationary platform GOES ABI and proposed an empirical bias correction scheme to improve the AOD accuracy. The GOES AOD product is potentially very useful in radiative forcing and air quality studies, in that it offers the diurnal variability of AOD on large scale. However, the existence of s bias in the diurnal cycle is a significant drawback that limits its use. Therefore, the bias correction scheme offered in this paper is both important and useful. However, I hope the authors can give more analysis proving and explaining that surface reflectance is responsible for the bias, and that the bias correction is effective under all AOD loading and surface conditions. These I think are major issues, although they should not be too difficult to address. My detailed comments are listed below.

Major comments:

1. I agree with the authors that surface reflectance parameterization is the most likely cause of the AOD bias. However, in the paper the authors seem very definitive on this point. For example, in the abstract, it says "ABI AOD has diurnally varying biases due to errors in the land surface reflectance relationship between the bands used in the ABI AOD retrieval algorithm". Therefore, I wonder if they can offer more detailed analysis proving this point and explain how the relationship between surface reflectance of different channels vary with geometry? The difference between the test position and the current operational position does not seem large enough to account for such high AOD bias. One possibility is that the NDVI also varies with solar zenith angle. Do the authors use MODIS NDVI? They are calculated from polar orbiting satellites and the NDVI only represent one solar zenith angle. Although NDVI should be a normalized quantity that is not affected by the angle, the large different solar position between polar orbit and geostationary orbits may cause MODIS NDVI not representative of all angles. 2. The bias correction assumes that the difference between 30-day minimum AOD and background AOD is the systematic error, and subtract this error from every AOD retrieval. I wonder if the bias also depends on AOD itself, i.e., aerosol loading, so that the systematic bias derived as above does not represent all AOD conditions? The validation set seems somewhat small (only 6 days of data) and all days have low AOD (<0.1). I thus wonder how the bias and correction algorithm may perform for high AOD cases? Another issue is that the effect of correction is not obvious for top quality data, mostly because the bias data are already removed from top quality (see Figure 1). Is this because these retrievals have high residual error so that they are removed from top quality set? Investigating the reason may offer some clue for the causes of the bias or algorithm improvements.

Minor comments:

1. Section 2.1: What cloud screening scheme is used? And which NDVI data is used, MODIS? 2. Section 2.2: Is there any quality control performed on AERONET Level

1.5 data? What is estimated AOD error? 3. Line 304, the following reference also points out the poor VIIRS aerosol model selection over China: 4. Comparison with PM2.5 seems not very relevant, and removing it does not impair the integrity of the study. There are a lot of factors affecting the AOD-PM2.5 relationship and I think this comparison may complicate the analysis. 5. Figure 6: could the authors also compare with MODIS to demonstrate the effect of bias correction? The peak of the bias happens at 17UTC, which is 1PM US east time and is close to Aqua overpass.

---

## Referee Comment (RC3) · Anonymous Referee #3 · 17 Apr 2020

"Improving GOES Advanced Baseline Imager (ABI) Aerosol Optical Depth (AOD) Retrievals using an Empirical Bias Correction Algorithm". Hai Zhang et al.

Submitted to AMT Discussions

Summary

Current operational retrievals of AOD from radiances measured by the ABI sensor on aboard of GOES 16 exhibit a diurnal bias (sun angle dependency) associated to the surface reflectance of the pixel under observation. This study introduces this problem and proposes an ad-hoc correction to the retrieved AOD. A correction is developed by collocating GOES AOD retrievals over selected Aeronet sites (mostly in the East

of USA). Only days with low and constant (through the day) Aeronet AOD values are used to ensure that the GOES deviations are caused by the solar angle changes and not from aerosol loading variations. The differences are then assessed and a correction based on those differences (which in turn are a function of geometry and NDVI) is created. The correction is assumed to be valid through the full ABI swath and then applied to retrieved AODs. The corrected AODs are validated against Aeronet during a 6-month period. The correction successfully improves the satellite-Aeronet AOD comparison. While the improvement is clear and it may result in a more accurate operational product, this analysis does not address the actual problem the causes the bias (a non-adequate surface reflectance data base) and presents an ad-hoc correction. In addition, I find that this study has important methodological defects and I do not recommend the paper for publication in this form.

——————————————————————————————————————-

Overall there are two major concerns about this work.

First, this a very empirical approach where the root of the problem is not addressed, namely the angular dependence of the surface reflectance as a function of sun angle. Although the authors do acknowledge that this is the real issue and they are working on it, they are content to use an ad-hoc approach by forcing the retrieved AOD to match the ground truth AOD. While this may be a reasonable practical correction, it does not show any new scientific approach (alternatively the authors do not highlight what is novel in doing this) and it does not attempt a correction on the actual measurement (observed radiances) based on physical principles (such as a modeled BRF) and using radiative transfer. With this regard, the work does not offer anything new.

Second, the validation is carried out by comparing the corrected retrievals against observations from the same instrument used for creating the correcting term. This is not adequate and it puts an asterisk on the goodness of the correction. At least these new corrected AODs need to be validated against an independent set of observations.

Also, note that in comparing figures 4c and 4d, there is a clear improvement in high AODs ($\sim$>0.5) whereas for lower AODs values, the scattering increases in figure 4d. This raises the question on whether the correction should be applied across the board to all aerosol loadings. This is relevant to AQ studies given that the vast majority of aerosol loadings are below AODs $\sim$< 0.5, it is very desirable to have those levels of loading well characterized.

It should be noted that this critique does not preclude or advise against the application this correction to the operational product. However, the material here presented does not have the depth required for a scientific report.

---

## Author Comment (AC2) · 26 May 2020

This paper evaluates the AOD retrieval from geostationary platform GOES ABI and proposed an empirical bias correction scheme to improve the AOD accuracy. The GOES AOD product is potentially very useful in radiative forcing and air quality studies, in that it offers the diurnal variability of AOD on large scale. However, the existence of bias in the diurnal cycle is a significant drawback that limits its use. Therefore, the bias correction scheme offered in this paper is both important and useful. However, I hope the authors can give more analysis proving and explaining that surface reflectance is responsible for the bias, and that the bias correction is effective under all AOD loading and surface conditions. These I think are major issues, although they should not be too difficult to address. My detailed comments are listed below.

**Major comments:**

I agree with the authors that surface reflectance parameterization is the most likely cause of the AOD bias. However, in the paper the authors seem very definitive on this point. For example, in the abstract, it says "ABI AOD has diurnally varying biases due to errors in the land surface reflectance relationship between the bands used in the ABI AOD retrieval algorithm". Therefore, I wonder if they can offer more detailed analysis proving this point and explain how the relationship between surface reflectance of different channels vary with geometry?

In the revised paper, we give a detailed case study at GSFC site for different geometries. Specifically, in the case study, the surface reflectance relationships used are closer to the real relationships in the afternoon than at noon, and therefore, the afternoon AODs retrieval are closer to the AERONET AODs.

The difference between the test position and the current operational position does not seem large enough to account for such high AOD bias.

We did not say it is the main reason. But it is one reason, although the effect may be small.

One possibility is that the NDVI also varies with solar zenith angle. Do the authors use MODIS NDVI? They are calculated from polar orbiting satellites and the NDVI only represent one solar zenith angle. Although NDVI should be a normalized quantity that is not affected by the angle, the large different solar position between polar orbit and geostationary orbits may cause MODIS NDVI not representative of all angles.

No, ABI AOD retrieval algorithm doesn't use MODIS NDVI. The algorithm uses ABI top of atmosphere reflectance of 0.64  $\mu$ m and 0.86  $\mu$ m bands to calculate it, independent from MODIS. NDVI is defined by red and NIR bands at TOA as

$$\text{NDVI} = \frac{\rho_{0.86}^{\text{TOA}} - \rho_{0.64}^{\text{TOA}}}{\rho_{0.86}^{\text{TOA}} + \rho_{0.64}^{\text{TOA}}}$$

The geometry dependence of NDVI is an issue, but not so large for the main cause of the AOD bias, as shown in the case study at GSFC in the revised paper. We are aware that this NDVI is not an "aerosol-resistant" NDVI. The choice of wavelength was dictated by the availability of ABI channels. MODIS AOD algorithm uses  $1.24 \mu m$  and  $2.12 \mu m$  band pair, i.e.

NDVI =  $\frac{\rho_{2.12}^{TOA} - \rho_{1.24}^{TOA}}{\rho_{2.12}^{TOA} + \rho_{1.24}^{TOA}}$ . However, ABI does not contain the 1.24 µm band. Based on the available ABI bands, we analyzed the dependence of 0.47 µm and 2.25 µm surface reflectance relationship to the NDVI from (0.64,0.86) µm pair and from (0.86,2.2) µm pair. It turned out that NDVI from (0.64,0.86) µm pair better separates the soil-based and vegetation-based. Based on this NDVI, the surface is classified into 4

different NDVI ranges and the surface reflectance parameterization are derived using ABI reflectances (Table 3-12 in ATBD), independent from MODIS.

**2. The bias correction assumes that the difference between 30-day minimum AOD and background AOD is the systematic error, and subtract this error from every AOD retrieval. I wonder if the bias also depends on AOD itself, i.e., aerosol loading, so that the systematic bias derived as above does not represent all AOD conditions?**

In the revised paper, an evaluation is performed for the bias correction algorithm for different AOD loading. Figure 7 in the revised paper shows the ABI AOD error and standard deviation in different AERONET AOD bins, with equal number of matchup data in each bin. For high quality AOD, bias correction reduces bias in the highest two AOD bins, with center around 0.3 and 0.57. In the range [0.1, 0.3], bias correction over corrects and introduces negative mean bias with slightly larger magnitude than the original mean bias, around 0.01 in magnitude differences. In the range [0,0.1], AOD mean biases are close to zero both before and after correction, but the bias correction AOD error has smaller standard deviation. For the top 2 qualities ABI AOD, bias correction reduces the bias in the whole AOD range with slight over corrections of magnitude of about 0.02 when AOD is greater than 0.1.

**The validation set seems somewhat small (only 6 days of data) and all days have low AOD (<0.1). I thus wonder how the bias and correction algorithm may perform for high AOD cases?**

Those are for case studies. The scatter plots in Figure 5 (in the revised paper) includes all the 5 months matchup data of AERONET sites over CONUS. Figure 7 (in the revised paper) and the corresponding discussion is also added to answer your question.

**Another issue is that the effect of correction is not obvious for top quality data, mostly because the bias data are already removed from top quality (see Figure 1). Is this because these retrievals have high residual error so that they are removed from top quality set? Investigating the reason may offer some clue for the causes of the bias or algorithm improvements.**

A lot of them are due to the relatively large standard deviation of 3x3 box in the 0.47 µm band, which is used in the ABI AOD retrieval algorithm to remove residual cloud contamination with a standard deviation threshold 0.006 for high quality AOD retrieval. This method was adopted from VIIRS retrieval. However, VIIRS retrieval uses a different band 0.41 µm, in which surface reflectance is much lower than 0.47 µm band. As a result, the standard deviation test likely erroneously removes clear pixels with high standard deviation caused by surface. Because the standard deviation information is only available in the intermediate product, which were not archived for long term use, we examined several granules of ABI AOD retrieval from off-line algorithm run and found that 65-80% in medium quality land pixels have standard deviation of 0.47 µm band above the threshold of 0.006.

**Minor comments:**

**1. Section 2.1: What cloud screening scheme is used?**

ABI has a cloud mask product (ABI Cloud Mask ATBD, 2012,

https://www.star.nesdis.noaa.gov/goesr/documents/ATBDs/Baseline/ATBD\_GOES-R\_Cloud\_Mask\_v3.0\_July%202012.pdf, last accessed 5/3/2020), which is used in the ABI AOD retrieval algorithm. In addition, several internal tests are performed to further remove contamination from cloud(ABI AOD ATBD, 2018) : (1) internal cloud test; (2) internal cirrus test; (3) internal inhomogeneity test.

**And which NDVI data is used, MODIS?**

No, not MODIS NDVI. The algorithm uses ABI top of atmosphere reflectance of 0.67  $\mu$ m and 0.86  $\mu$ m bands to calculate it: NDVI is defined by red and NIR bands at TOA as

NDVI =
$$\frac{\rho_{0.86}^{TOA} - \rho_{0.64}^{TOA}}{\rho_{0.86}^{TOA} + \rho_{0.64}^{TOA}}$$

**2. Section 2.2: Is there any quality control performed on AERONET Level 1.5 data? What is estimated AOD error?**

Level 1.5 AERONET AOD data is cloud screened and quality controlled, with a + 0.02 bias and one sigma uncertainty of 0.02 (Giles et al., 2019).

**3.** Line 304, the following reference also points out the poor VIIRS aerosol model selection over China:**

Thanks. We included this reference in the paper

**4. Comparison with PM2.5 seems not very relevant, and removing it does not impair the integrity of the study. There are a lot of factors affecting the AOD-PM2.5 relationship and I think this comparison may complicate the analysis.**

One of the main applications of NOAA AOD product is to operationally derive surface PM2.5 for air quality monitoring and forecasting applications. Arguably, there are many factors that affect the AOD to PM2.5 relationship (aerosol composition, aerosol layer height, relative humidity, time of observation, accuracy of AOD, etc.). It is intuitive that an accurate AOD gives a better estimate of surface PM2.5 given that other factors influencing this relationship the way they are. Therefore, demonstrating that the relationship improves with improved AOD is quite important for our studies and work we do with user community.

**5. Figure 6: could the authors also compare with MODIS to demonstrate the effect of bias correction? The peak of the bias happens at 17UTC, which is 1PM US east time and is close to Aqua overpass.**

The MODIS AOD from Aqua dark target and deep blue algorithm are added (Figure 9 in the revised paper). The bias corrected ABI AOD compares very well with deep blue MODIS AOD in both magnitude and data coverage.

**Reference**

Giles, D. M., Sinyuk, A., Sorokin, M. G., Schafer, J. S., Smirnov, A., Slutsker, I., Eck, T. F., Holben, B. N., Lewis, J. R., Campbell, J. R., Welton, E. J., Korkin, S. V., and Lyapustin, A. I.: Advancements in the Aerosol Robotic Network (AERONET) Version 3 database – automated near-real-time quality control algorithm with improved cloud screening for Sun photometer aerosol optical depth (AOD) measurements, Atmos. Meas. Tech., 12, 169–209, https://doi.org/10.5194/amt-12-169-2019, 2019.

---

## Author Comment (AC3) · 26 May 2020

**"Improving GOES Advanced Baseline Imager (ABI) Aerosol Optical Depth (AOD) Retrievals using an Empirical Bias Correction Algorithm". Hai Zhang et al.**

**Submitted to AMT Discussions**

**Summary**

**Current operational retrievals of AOD from radiances measured by the ABI sensor on aboard of GOES 16 exhibit a diurnal bias (sun angle dependency) associated to the surface reflectance of the pixel under observation. This study introduces this problem and proposes an ad-hoc correction to the retrieved AOD. A correction is developed by collocating GOES AOD retrievals over selected Aeronet sites (mostly in the East of USA). Only days with low and constant (through the day) Aeronet AOD values are used to ensure that the GOES deviations are caused by the solar angle changes and not from aerosol loading variations. The differences are then assessed and a correction based on those differences (which in turn are a function of geometry and NDVI) is created. The correction is assumed to be valid through the full ABI swath and then applied to retrieved AODs. The corrected AODs are validated against Aeronet during a 6-month period. The correction successfully improves the satellite-Aeronet AOD comparison. While the improvement is clear and it may result in a more accurate operational product, this analysis does not address the actual problem the causes the bias (a non-adequate surface reflectance data base) and presents an ad-hoc correction. In addition, I find that this study has important methodological defects and I do not recommend the paper for publication in this form.**
* * *
**Overall there are two major concerns about this work.**

**First, this a very empirical approach where the root of the problem is not addressed, namely the angular dependence of the surface reflectance as a function of sun angle. Although the authors do acknowledge that this is the real issue and they are working on it, they are content to use an ad-hoc approach by forcing the retrieved AOD to match the ground truth AOD. While this may be a reasonable practical correction, it does not show any new scientific approach (alternatively the authors do not highlight what is novel in doing this) and it does not attempt a correction on the actual measurement (observed radiances) based on physical principles (such as a modeled BRF) and using radiative transfer. With this regard, the work does not offer anything new.**

This is an approach in addition to the traditional approach based on physical principles. The uncertainty in BRF model is transferred to AOD. Improving AOD is the same as improving BRF. The approach in this paper solves the problem in AOD space instead of BRF space, which is different from traditional approach. The traditional approach uses AERONET matchup dataset to generate surface reflectance relationships and then assume these relationships can also be applied to surfaces at other places where AERONET stations are not present. Even at the AERONET sites, the surface reflectance relationships have large uncertainty. The bias correction algorithm proposed here can reduce those uncertainties. More importantly, it does not rely on AERONET surface and therefore can be applied anywhere else without assuming everywhere else is the same as AERONET.

The empirical bias corrections to retrieved AODs is not new. The NASA MODIS Dark Target AOD algorithm corrects AOD using a bias correction algorithm over urban areas using post processing of AODs for areas where urban land percentage is greater than 20% (Gupta et al., 2016). There are other MODIS AOD correction algorithms as well developed by users for their own applications (e.g., Lary et

al., 2009).  In fact, compared to these bias correction algorithms, our approach is better because it is internally consistent and does not rely on any external dataset.  Moreover, the bias correction preserves the original AOD data file, and it is "self-correcting", meaning if the physical AOD algorithm improves the bias correction will automatically adjust to the new values.

**Second, the validation is carried out by comparing the corrected retrievals against observations from the same instrument used for creating the correcting term. This is not adequate and it puts an asterisk on the goodness of the correction. At least these new corrected AODs need to be validated against an independent set of observations.**

The comparison is between the correction before correction and after correction to show the improvement.  The 30-day AOD used is assumed to contain information of the bias.  For most of the days of validation, except the first 30 days, we use the 30-day period data before the day to get the AOD bias, which is independent from the data being corrected.  The following figure shows the scatter plots of the validation using the data with the first 30 day removed, and therefore the bias corrected ABI AOD data are totally independent from the data used for obtaining the AOD bias.  The conclusions remain the same as in the paper.

[Figure]

Figure A.  Scatter plots of GOES-16 ABI AOD vs AERONET AOD for September 5, 2018 to December 31, 2018 across the CONUS domain: (a) high quality ABI AOD before bias correction, (b) top 2 qualities ABI AOD before bias correction, (c) high quality ABI AOD after bias correction, and (d) top 2 qualities ABI AOD after bias correction. In the plots, N is the number of matchups, R is the correlation coefficient, and RMSE is the root mean square error.

**Also, note that in comparing figures 4c and 4d, there is a clear improvement in high AODs (∼>0.5) whereas for lower AODs values, the scattering increases in figure 4d.**

Figure 4c and 4d are both after correction AOD. One is high quality and the other is high and medium quality. Figure 4d has more data points than 4c and they don't have one-to-one correspondence. Therefore, they should not be compared for improvement of bias correction. The correct comparison is between Figure 4a and Figure 4c, and between Figure 4b and Figure 4d. (Figure 4 changes to Figure 5 in the revised paper).

**This raises the question on whether the correction should be applied across the board to all aerosol loadings. This is relevant to AQ studies given that the vast majority of aerosol loadings are below AODs ∼< 0.5, it is very desirable to have those levels of loading well characterized.**

We plotted comparisons of AOD errors vs AERONET AOD for different AOD loadings in Figure 7 in the revised paper. The bias corrected AOD are shown to have reduced bias for all the AOD ranges for the top 2 (high and medium) qualities AOD. For high quality AOD, bias correction reduces bias in the highest two AOD bins, with center around 0.3 and 0.57. In the range [0.1, 0.3], bias correction over corrects and introduces negative mean bias with slightly larger magnitude than the original mean bias, around 0.01 in magnitude differences. In the range [0,0.1], AOD mean biases are close to zero both before and after correction, but the bias correction AOD error has smaller standard deviation.

**It should be noted that this critique does not preclude or advise against the application this correction to the operational product. However, the material here presented does not have the depth required for a scientific report.**

The algorithm is an effective tool to evaluate and correct the AOD bias from geostationary satellites. If you agree that the algorithm works, we should have it published so the other researchers can benefit from the improvements and use this data in their studies/applications. For example, apply it on AOD product from other geostationary satellite platform or other retrieval algorithm.

We hope the new results added to the revised paper also adds more depth to the material presented. We believe that even though an empirical "technique" for correcting biases in a specific product is presented, the approach and its evaluation presented do have merits in its application to other AOD products as well, and thus others could benefit from its publication.

References

Gupta, P., Levy, R. C., Mattoo, S., Remer, L. A., and Munchak, L. A.: A surface reflectance scheme for retrieving aerosol optical depth over urban surfaces in MODIS Dark Target retrieval algorithm, Atmos. Meas. Tech., 9, 3293–3308, https://doi.org/10.5194/amt-9-3293-2016, 2016.

D. J. Lary, L. A. Remer, D. MacNeill, B. Roscoe and S. Paradise, "Machine Learning and Bias Correction of MODIS Aerosol Optical Depth," in IEEE Geoscience and Remote Sensing Letters, vol. 6, no. 4, pp. 694-698, Oct. 2009, doi: 10.1109/LGRS.2009.2023605.

---

## Author Comment (AC1)

*The goal of this paper is to correct for AOD retrieval biases in GOES ABI AOD product using an empirical approach. The surface reflectance in the current GOES AOD algorithm is estimated based the relationship between 0.47 and 2.2 um and 0.64 and 2.2 um since most aerosols are 'transparent' in the 2.2 um. This is based on the Kaufman et al (1997, IEEE) paper that many MODIS algorithms use to estimate surface reflectance. In this paper, the authors look for 'clear days' (based on AERONET AOD values less than 0.05) to assess the GOES AOD (for both high and medium quality retrievals) for a few selected sites. They report that the GOES AOD is biased since the GOES AOD is much larger than the AERONET AOD for these clear days. The authors note that the biases appear to be centered around 1700 UTC and it is due to surface reflectance parametrizations at various sun-satellite viewing geometries. The authors then attempt to correct this bias based on the premise that it is the surface reflectance that is the issue in the GOES algorithm. Then they use a 30-day composite of GOES AOD to estimate the minimum AOD and subtract that with the background AOD (a fixed value of .025) to correct for the bias. They use two polynomial fitted relationships to estimate biases. They then correct the AOD using these relationships and then validate the results with AERONET AOD and show improvement in these biases.*

*First I need to note that the paper needs to go through some editorial clean up since several sentences are awkward; key references (Kondragunta et al 2020) are missing;and some references are really old.*

We removed several old references and checked the references.

*I find several problems with the paper and most importantly it is the use of AOD to make these corrections rather than working with the reflectances. The algorithm retrieves AOD based on apriori assumptions of aerosol model, surface parametrizations based on NDVI , cloud clearing approaches and a host of thresholds for cloud cover, inhomogeneity, etc (ATBD, 2018). Now this paper indicates that the surface parametrizations are a problem and then to remove the biases the authors use the retrieved AOD to make bias adjustments. The original algorithm uses reflectance ratios to arrive at surface values and now this paper goes back to the older GASP approach to obtain the 30-day composite minimum (not reflectance) AOD values. Looking at this from an algorithm perspective it is not the correct solution for an operational algorithm to go through retrieval using one set of processes, retrieve AOD's and then use the retrieved AOD values to make corrections for parameters that are part of the original retrieval process (in this case surface reflectance). The authors need to think about having the correct algorithm as part of the retrieval process rather than adjusting it after the retrieval is done.*

The purpose of the bias correction algorithm is to correct the bias in an already existing aerosol optical depth product.  It is not intended to substitute for the original AOD algorithm. We agree, and has been fully aware of, that ideally reduction of biases should be dealt with in the AOD algorithm itself. As discussed in the paper, the deviation of the real spectral surface reflectance relationships from the parameterization used in the retrieval can cause the AOD retrieval bias.  Improving the spectral surface reflectance relationships is the subject of an independent, parallel work, and thus it is not discussed in the current paper.  Once such an improvement becomes available and is shown to satisfactorily reduce the AOD bias, the bias correction may be turned off.  But before that happens, we plan using the bias correction algorithm to provide users an AOD product with improved accuracy and coverage.  On the other hand, surface reflectance relationship parameterization is derived at AERONET sites and is assumed to be valid over all other areas.  This assumption may not hold everywhere.  Actually there are very few evaluations over areas other than AERONET sites.  The bias correction algorithm can evaluate AOD bias over areas other than AERONET sites and reduce the bias there.  The empirical bias corrections to retrieved AODs is not new.  The NASA MODIS Dark Target AOD algorithm corrects

AOD using a bias correction algorithm over urban areas using post processing of AODs for areas where urban land percentage is greater than 20% (Gupta et al., Atmos. Meas. Tech., 9, 3293-3308, 2016). There are other MODIS AOD correction algorithms as well developed by users for their own applications (e.g., Lary et al. 2009). In fact, compared to these bias correction algorithms, our approach is better because it is internally consistent and does not rely on any external dataset.

***The other issue is the relaxation of quality flags to allow more data. There were strong reasons for picking all the metrics for high and medium quality flags in the first place (ATBD, 2018) whether it is cloud/snow cover or inhomogeneity. Line 90 to 95 provides the various reasons for selecting the pixels for the retrievals and this paper now allows all the medium quality flags in the process but does not address cloud contamination issues.***

There is always a tradeoff between better data coverage and reducing cloud contamination. From the scatter plots in Figure 4 (Figure 5 in the revised paper), the bias corrected high and medium qualities AODs have statistics close to that of the high quality alone, which suggests that once the bias correction is applied we can use data that were assigned either high or medium quality data in the original AOD without sacrificing accuracy. In other words, the expectation is that the AOD at pixels that are subjected to potential cloud/snow contamination, and thus are labeled as medium quality, are corrected (at least partly) for this contamination as a result of bias correction.

***The paper needs to be more convincing that it is indeed surface issues and not cloud cover that causes these problems. The results need to be discussed in terms of scattering angles (see She et al, Remote Sensing, 2019). This will allow more quantitative analysis rather than statements like those in 160-161.***

We adopted your suggestion and plotted the scattering angle dependence of the error, comparing the AOD errors before and after correction. The original ABI AOD errors have a scattering angle dependence in the plots. After applying the bias correction algorithm, the scattering angle dependence of the bias is reduced. (Figure 6 in the revised paper and corresponding discussions).

There are several reasons that the bias is not caused by cloud contamination: (1) the diurnal pattern of retrieved ABI AOD on clear days always has a peak at around noon and the peak gradually reduces away from noon; cloud contamination is not expected to produce such a pattern; (2) cloud contaminations are random errors instead of systematic errors shown in the paper. Random errors from cloud contamination won't be corrected by our algorithm. The effectiveness of our algorithm in removing systematic errors indicates that the main reason of the bias is not cloud contamination. (3) If in some cases cloudiness at a given location has its own diurnal cycle and introduces a systematic bias, the bias correction corrects it too. The bias correction algorithm does not differentiate where the bias comes from and it corrects the bias as long as the bias is systematic.

***Also for Figure 1 and Figure 2 what were the histograms of actual reflectance's from the GOES channels for the various peaks. This can help explain Figure 2 better.***

Instead of histogram of the surface reflectances, we plotted scatter plots of the 0.47 µm and 2.2 µm surface reflectances, surface reflectance relationship used in the ABI AOD retrieval algorithm, and the histograms of NDVI for six observations around the GSFC site (Figure 3 in the revised paper). The analysis shows that the surface reflectance relationship used in the retrieval algorithm is directly connected to the ABI AOD retrieval biases. The change of the peak ABI AOD bias amplitude is related to the different relationship used because of the differences in NDVI in the three days.

*The paper uses two sets of parametrizations for adjusting the biases and then in line 255 back tracks the approach by stating that this could have large uncertainties.*

We don't expect a parameterization to fit every situation. As long as it works for the majority of the locations and/or geometries, it can be used. Notice that even for the worst case in the early morning, for the University of Houston, the peak bias is reduced from 0.4 to 0.3 (Figure 1e and Figure 8e in the revised paper).

*Figure 8 and Figure 9 appears as a complete afterthought since the aerosol model discussion is not complete or convincing.*

We respectfully disagree with the reviewer that this is an afterthought. One of the challenges of aerosol remote sensing is the representation of aerosol optical and physical properties in the models used to generate Look-up-Tables for retrievals. Aerosol model selection over land is a known problem in MODIS/VIIRS type sensors. We don't expect to be able to solve it with the bias correction algorithm either. Here we just want to point out that the problem exists.

*I have no idea why the PM2.5 discussions (Figure 9) is relevant for this paper.*

One of the main reasons why NOAA generates near real time AOD retrievals is for user applications related to air quality monitoring and forecasting. Users use AOD as a proxy for surface PM2.5 and among many things that impact this relationship, accuracy of AOD itself is very important. Better AOD retrieval means better PM2.5 estimates from satellite, which is an important application of satellite AOD product.

*The AERONET data used is from 2018 and the authors need to be using Level 2 not 1.5. This data should be available.*

Some sites still don't have Level 2 data yet. For example, as of April 20 2020, GSFC still does not have level 2 data available for the days after September 2018. In our daily work, we routinely do our analysis with both Level 2 and Level 1.5 data and we are quite comfortable in using Level 1.5 data.

*Other issues. Define accuracy and precision and be quantitative rather than merely stating that one product is better than the other.*

We added the corresponding numbers into the places where we discuss the accuracy and precision.

*Line 50, Deemed to have quality sufficient is rather vague.*

Sentence removed.

*Line 73: The word transparent to most aerosols is rather vague. Describe why this is possible briefly based on aerosol size and extinction*

We removed this sentence. This is the assumption of the original MODIS algorithm. It was abandoned later on so that the retrieval is more accurate. 2.2 µm band is approximately transparent to small sized particles such as smoke, urban aerosols, but it is not as transparent to large particles such as dust. The extinction is determined by the ratio between the wavelength and the particle size. Based on Mie theory, the larger the ratio, the smaller the extinction.

*Line 74. Again, poor phrasing. It is not linear reflectance BETWEEN channels if it is three channels. Be specific.*

Changed to "The algorithm assumes linear relationships exist between the surface reflectance of 0.47 µm band and 2.2 µm band, and between those of 0.64 µm band and 2.2 µm band."

*Line 75-80 is awkward phrasing. The algorithm does not make retrievals? Describe the algorithm clearly but briefly. Line 81-84 is not clear at all.*

*While I understand how the algorithm works this type of writing will not help all readers understand the algorithm and methods used in this paper.*

Revised the paragraph as follows.

Over land, three ABI channels are used in the retrieval, i.e. 0.47 µm, 0.64 µm, and 2.2 µm. The algorithm assumes linear relationships exist between the surface reflectance of 0.47 µm band and 2.2 µm band, and between 0.64 µm band and 2.2 µm band. The coefficients of the relationships are functions of NDVI (between 0.86 and 0.64 µm channel) and solar zenith angle (GOES-R ABI AOD ATBD, 2018). Other atmospheric and geographic parameters needed for the retrieval are also inputted, such as surface pressure, surface height, total column ozone, etc. The algorithm only retrieves AOD over dark surface, when the TOA reflectance in the 2.2 µm band is less than 0.25. The retrieval algorithm contains two steps. In the first step, one of four aerosol models is assumed, i.e. dust, smoke, urban, and generic, and AOD for each of the aerosol model is retrieved using the 0.47 µm and the 2.2 µm bands. The algorithm uses a Look-up-Table (LUT) to perform radiative transfer calculation. The LUT stores reflectances, transmittances and other quantities for discrete states of atmosphere and Sun-satellite geometries. For each AOD in the LUT, the algorithm performs atmospheric correction in 2.2 µm band to obtain surface reflectance in that band, and uses the 0.47 µm and the 2.2 µm band relationship to obtain 0.47 µm band surface reflectance. TOA reflectance in the 0.47 µm band can then be calculated using the LUT. The AOD for the assumed aerosol model is obtained through interpolation of the two AODs that give TOA reflectances in the 0.47 µm band closest to the satellite measurement. At the end of this step, there are four AOD solutions from the 0.47 µm band and 2.2 µm band, one for each aerosol model. In the second step, one of the four solutions is then selected as the final retrieval using the 0.64 µm channel by looking for the aerosol model that gives a TOA reflectance in that channel that is the closest to the observed TOA reflectance. In this step, 0.64 µm band TOA reflectance is calculated with 2.2 µm band surface reflectance from last step, relationship between 0.64 µm band and 2.2 µm band and AOD of corresponding aerosol model. The algorithm does not make retrievals over bright land pixels, pixels covered by cloud or snow, etc. The AOD retrieval range is [-0.05,5] and any retrievals greater than 5 are marked as out of range.

*Line 89: Usually very small? What does that mean? Need some numbers.*

Added: "For example, the ratio between the number of the top 2 qualities and the high quality matchup with AERONET is about 2 (see the following section), while the ratio is 1.2 for VIIRS AOD (Laszlo and Liu, 2016). "

*Lines 89-94 needs to be clearer with brief discussion rather than listing the problems.*

They are just a list of criteria used to degrade AOD quality in the current algorithm. The starting sentence was revised as: "Following criteria are used to degrade a pixel from high quality to medium quality: …"

The problem is the standard deviation test. We did discuss it in the next sentences.

The reasons for the other criteria are out of the scope of this paper and are not discussed in the paper. Following are the reasons about the cloud/snow adjacency criteria:

Pixels close to clouds or snow can be potentially impacted by radiation scattered from them into the cloud-free and snow-free columns (e.g. Marshak and Davis, 2005; Lyapustin and Kaufman, 2001). For clouds, there is also the issue of transition from clear to cloudy, which is gradual. Cloud detection may not label these pixels as cloudy because they are not bright enough. At the same time these pixels have cloud droplets mixed with aerosol, and/or a humidity that results in aerosols, if they are hygroscopic, which are not well represented by any of the models in the LUT (e.g. Jia et al., 2019; Tang et al., 2019) .

***Line 98: If the surface reflectance issues are so different between 0.41 and 0.47 micron then the authors need to show or discuss this for certain land types. Otherwise these statements are vague.***

Added: Over CONUS region, from VIIRS data, the 0.41 µm surface reflectance is 0.3-0.4 times the 0.67 µm band surface reflectance and the 0.47 µm surface reflectance is 0.5-0.6 times the 0.67 µm surface reflectance (Zhang et al., 2016).  Therefore, 0.41 µm surface reflectance is 20%-50% lower than 0.47 µm surface reflectance.

***115-120 discussion is not "technical" enough. What does air mass movements mean? You need to then state what wind speeds at what height provide the 27.5 km radius.***

We removed "the air mass movements" in the sentence. For this matchup, we did not do temporal matchup with AERONET and just plotted the time series of ABI AOD and AERONET AOD.  To our knowledge, the air mass movements argument first appeared in Ichoku et al. (2002) as follows: " the average travel speed of an aerosol front is of the order of 50 Km/h. This was visually estimated from animated daily sequences of TOMS aerosol index images (http://jwocky.gsfc.nasa.gov/aerosols/aermovie.html) for July to September 1988, where aerosol fronts are seen crossing the Atlantic from the west coast of Africa to the East coas t of America (approximately 6000 Km) in about five or six days. Therefore, the 50x50 Km window would match a 1-hour sunphotometer data segment. All references to MODIS spatial statistics in the rest of this paper imply those based on the 50x50 Km (5x5 pixel) subset grid".  They did not mention the height of the aerosol layer.

***Line 140+: How about retrieval biases due to sun-satellite viewing geometry in radiative transfer code?***

We are not aware of any report in the literature of AOD retrieval errors with magnitude ≥ 0.1 due to radiative transfer model within the range of ABI AOD retrieval geometry. Errors may be present at the edge of the disk due to plane parallel assumption but those retrievals are not recommended for even qualitative use, and they were excluded from the current analysis.

***Line 147: We need to see these relationships between two channels for the solar geometries.***

They are added in the paragraph of case studies at GSFC site.

***I find the two reasons in 152-155 to be problematic. Why should the test position issue matter if these relationships are established for certain solar viewing geometries/NDVI?***

The parameterization is a simplified model that assumes the relationships depend only on solar zenith angle and NDVI.  However, in reality, the relationships depend on all the angles, i.e. solar zenith angle, satellite zenith angle, solar azimuthal angle and satellite azimuthal angle, and surface type (not only NDVI).  In addition, NDVI is also a function of those angles.  When the satellite moved, the satellite angles changed.  Unless the relationships and NDVI are independent of satellite angles, the relationships should change.

*Plus there are reasons why the quality flags were established for high, low, medium in the first place (cloud cover, snow cover etc). Of course one would use the best quality flags for establishing surface reflectance relationships because of contamination issues. Now if you are using medium quality flags to get more data into the analysis then of course your surface reflectance relationships are going to be different.*

This is the problem of surface reflectance relationships parameterization: they cannot be generalized to other pixels without losing AOD retrieval accuracy. With bias correction, we can correct those biases caused by this problem.

*Since this paper is about surface reflectance issues the authors need to show these relationships that currently exist for various angles/NDVI first to make their case stronger.*

We understand what you are saying but we think the paper is not about surface reflectance issues. It is about correcting the bias that, we think, happened to be caused primarily by deficiencies in the way we parameterize the relationship between spectral surface reflectances. The detailed surface relationships are available in GOES-R ABI AOD ATBD (2018) and is out of the scope of the paper.

But for your information, following is a summary of the relationships:

The surface reflectance relationships used in the above retrieval algorithm are derived through studies of ABI pixels near AERONET sites, where AODs are accurately measured from the ground and are considered as ground truth. A set of stringent pixel selection rules are applied to build a matchup dataset between ABI pixels and AERONET AOD in order to reduce cloud contamination and uncertainties in aerosol models (GOES-R ABI AOD ATBD, 2018). If AERONET AOD is less than 0.2 of a matchup dataset, surface reflectance of the pixels at the three channels are retrieved through atmospheric correction. With surface reflectance of all such pixels, the relationships are then derived and parameterized as functions of the solar zenith angle for different ranges of the normalized difference vegetation index (NDVI, between 0.86 and 0.64 µm channel) through linear regression analysis of the spectral surface reflectance. The current surface reflectance relationships are derived from ABI full disk matchup dataset in the time period of 04/29/2017 – 01/15/2018.

The surface reflectance relationships obtained are described in the following equations:

$$\rho_{0.47}[\rho_{0.64}] = (c_1 + c_2\theta_s) + (c_3 + c_4\theta_s)\rho_{2.2} \tag{1}$$

Where $\rho_{0.47}$, $\rho_{0.64}$, $\rho_{2.2}$ are surface reflectance at the three bands, $c_1, c_2, c_3, c_4$ are constants depending on NDVI as shown in Table 3-12 of the ATBD (shown in the following), $\theta_s$ is the solar zenith angle. NDVI is defined by red (0.67 µm) and NIR (0.86 µm) bands at TOA as

$$NDVI = \frac{\rho_{0.86}^{TOA} - \rho_{0.64}^{TOA}}{\rho_{0.86}^{TOA} + \rho_{0.64}^{TOA}} . \tag{2}$$

Table 3-12. Coefficients in the spectral surface reflectance relationship for different ranges of NDVI.

| Channels (μm) | $c_1$ | $c_2$ | $c_3$ | $c_4$ |
|---|---|---|---|---|
| | $NDVI \geq 0.55$ | | | |
| 0.47 vs. 2.25 | 1.436330E-02 | 2.060893E-04 | 1.749239E-01 | -2.859502E-03 |
| 0.64 vs. 2.25 | 1.374160E-02 | -5.128175E-05 | 2.761044E-01 | 1.034823E-03 |
| | $0.3 \leq NDVI < 0.55$ | | | |
| 0.47 vs. 2.25 | 4.163894E-02 | -2.147513E-04 | 1.598440E-01 | 7.401292E-04 |
| 0.64 vs. 2.25 | 2.990101E-02 | -1.873911E-04 | 4.602174E-01 | 9.658934E-04 |
| | $0.2 \leq NDVI < 0.3$ | | | |
| 0.47 vs. 2.25 | 5.154307E-02 | 5.679386E-05 | 2.048702E-01 | -7.064656E-04 |
| 0.64 vs. 2.25 | 5.179930E-02 | -1.043257E-04 | 4.937035E-01 | 4.310074E-04 |
| | $NDVI < 0.2$ | | | |
| 0.47 vs. 2.25 | -4.990575E-02 | 2.138207E-03 | 8.498076E-01 | -1.179596E-02 |
| 0.64 vs. 2.25 | -3.397737E-02 | 1.640336E-03 | 1.087497E+00 | -9.538776E-03 |

In the revised paper, we provide a detailed analysis and the surface reflectance relationships used over the GSFC site.

***The authors should also show the reflectance values on these plots so we can interpret the results better.***

We added them in the surface reflectance discussion for GSFC case study.

References

GOES-R Advanced Baseline Imager (ABI) Algorithm Theoretical Basis Document For Suspended Matter/Aerosol Optical Depth and Aerosol Size Parameter, NOAA/NESDIS/STAR, Version 4.2, February 14, 2018, https://www.star.nesdis.noaa.gov/smcd/spb/aq/AerosolWatch/docs/GOES-R_ABI_AOD_ATBD_V4.2_20180214.pdf, accessed 02/24/2020.

Gupta, P., Levy, R. C., Mattoo, S., Remer, L. A., and Munchak, L. A.: A surface reflectance scheme for retrieving aerosol optical depth over urban surfaces in MODIS Dark Target retrieval algorithm, Atmos. Meas. Tech., 9, 3293–3308, https://doi.org/10.5194/amt-9-3293-2016, 2016.

Ichoku, C., Chu, D.A., Mattoo, S., Kaufman, Y.J., Remer, L.A., Tanré, D., Slutsker, I. and Holben, B.N.: A spatio-temporal approach for global validation and analysis of MODIS aerosol products, Geophys. Res. Lett., 29(12), 8006, doi:10.1029/2001GL013206, 2002.

Jia, H., Ma, X., Quaas, J., Yin, Y., and Qiu, T.: Is positive correlation between cloud droplet effective radius and aerosol optical depth over land due to retrieval artifacts or real physical processes?, Atmos. Chem. Phys., 19, 8879–8896, https://doi.org/10.5194/acp-19-8879-2019, 2019.

D. J. Lary, L. A. Remer, D. MacNeill, B. Roscoe and S. Paradise, "Machine Learning and Bias Correction of MODIS Aerosol Optical Depth," in IEEE Geoscience and Remote Sensing Letters, vol. 6, no. 4, pp. 694-698, Oct. 2009, doi: 10.1109/LGRS.2009.2023605.

Lyapustin, A. I. and Kaufman, Y. J.: Role of adjacency effect in the remote sensing of aerosol, J. Geophys. Res., 106, 909–916, 2001.

Marshak, A., and Davis, A. (Eds), 3D Radiative Transfer in Cloudy Atmospheres, Springer Science & Business Media, 2005.

Tang, M., Chan, C. K., Li, Y. J., Su, H., Ma, Q., Wu, Z., Zhang, G., Wang, Z., Ge, M., Hu, M., He, H., and Wang, X.: A review of experimental techniques for aerosol hygroscopicity studies, Atmos. Chem. Phys., 19, 12631–12686, https://doi.org/10.5194/acp-19-12631-2019, 2019.

---

## Author Response (AR2)

[revised manuscript text omitted]

Dear Andy and anonymous reviewers,

Thank you for the thorough second review of our manuscript. We have carried out several analyses and theoretical calculations to address the concerns you raise. Details are given below as responses to individual editor/reviewer comments but here is a high-level summary of the work we did to address the comments:

(1) We obtained one month of GOES-16 ABI AOD retrievals derived using NASA MODIS Dark Target (DT) algorithm from Rob Levy and Pawan Gupta (personal communication) and applied our bias correction algorithm. We demonstrate that NASA DT ABI AOD also has a diurnal bias and the bias is reduced by using our bias correction algorithm;

(2) We added a new section 6 to discuss surface reflectance errors. We show how the parameterization for spectral surface reflectance relationship between 0.47 µm and 2.2 µm that is dependent on solar zenith angle and NDVI developed using data for a certain time period and geographical coverage can lead to errors when applied to a different time period and different geographical coverage. These errors translate into errors in AOD. We note that when we developed the parameterization as a function of solar zenith angle, we found the fitting error has little dependence on the scattering angle. Since the view zenith angles are fixed for geostationary satellites, we were hoping an explicit function of solar zenith angle would help better capture the diurnal variation. As you suggested to include scattering angle in the surface relationship, our on-going efforts on updating the surface reflectance relationship did find a parametrization dependent on solar and view zenith angles, and scattering angle could better capture the overall diurnal variation of AOD. This conclusion was reached in a series of empirical exercises, the description of which is beyond the scope of this paper.

(3) We also conducted radiative transfer simulations to demonstrate that error in the estimated surface reflectance and incorrect aerosol model selection are the two dominating sources for AOD bias with surface reflectance being the main source, especially when AODs are low.

(4) We also conclude that while updating the parameterization frequently to minimize the AOD bias is an option, it is not practical. Even if it is done, biases will remain because the parameterization will never be 100% accurate. This in fact is, and will be, a problem for any similar aerosol retrieval algorithm. Until now the aerosol remote sensing community spent time understanding polar-orbiting satellite geometries and retrieval accuracies, and the community has just begun to work with the geostationary satellite measurements. The one paper published by Gupta et al. (AMT, 2019) on Himawari-8 AOD retrievals using DT algorithm also shows diurnal bias but the authors have not probed the causes of that bias.

Given below are our responses (normal font) to individual reviewer comments (bold font).

**One of the reviewer criticisms of the original work was limited scope: a post-retrieval correction to an operational retrieval scheme, rather than an improvement to the algorithm, or something which could be applied more broadly. Although I understand from an operational point of view why this would be done, reviewers questioned whether this is something which warrants a journal publication, or would be better served as a technical report. One reviewer feels the same about the revised version, as the level of technical detail doesn't provide a direct path to broader applicability or to improving the algorithm in the future. I lean in this direction. The bias correction is in essence a quadratic fit to AOD compared to a reference point taken as unbiased. That is not a particularly**

**detailed approach: it's great it works, but that doesn't mean it's got enough broad interest, applicability, or detailed physics shown for a publication. The reviewer's comments are reproduced here:**

We tested the general applicability of the bias correction algorithm on an independent algorithm, the MODIS Dark Target (MODIS DT) algorithm, run on one month of GOES-16 ABI L1B data. The MODIS DT ABI AOD product was obtained from NASA. That product is at 10 km resolution and for the full disk. The results show that the MODIS DT AOD retrievals also exhibit similar bias pattern. The NOAA bias correction algorithm, when applied to the GOES-16 ABI NASA DT AOD, effectively reduced the bias. The results and additional analysis in the revised paper also indicate that the bias issue is inherent to the DT retrieval approach and needs to be addressed. The bias correction algorithm reduces the AOD bias at pixel level. Because the surface reflectance relationship is derived from an ensemble of pixels representing reflectances from a large variety of surfaces, even if it was improved inside the retrieval algorithm, it would not exactly represent the true relationship for a particular pixel and for a particular situation (time of the day, season, surface type). In the revised paper, we use radiative transfer simulations to demonstrate that the empirical bias correction is similar to fixing surface reflectance bias.

**"For the sake of brevity, I will address the first response of the author as I think the other questions are more technical and not so defining of the nature of this study.**

**The issue I raise is that I see very little value in a study that limits itself to obtain a correction without addressing the underlaying causes. This study provides a fix to a remotely sensed product using a methodology that seems to work but does not shed any meaningful light on nature of the problem causing the discrepancies first noted. While the methodology seems reasonable, the study does not stand out above a technical report offering a nudge to fix to an operational product.**

**I acknowledge that in an operational setting and near real time situations, speed is important and as long as the end product is satisfactory, ends justify the means to achieve the desired result. However, for scientific applications the correspondence between end product (in this case AOD) and connection with the physical process in the modeled retrieval is essential. This is the weakness of this study and it fails to make the point why this is of scientific interest if there are not causative demonstrations of the problem such as radiative transfer study, exploration of different surface databases or comparisons with other satellites. As written this study reads as technical report addressing a correction needed because the end product does not match well with independent observations.**

This is an approach in addition to the traditional approach based on physical principles because the uncertainty in Bi-directional Reflectance Function (BRF) model is transferred to AOD; improving AOD is the same as improving BRF. The approach in this paper solves the problem in AOD space instead of BRF space, which is different from the traditional approach. The traditional approach uses a satellite-AERONET matchup dataset to generate spectral surface reflectance relationships and then assumes these relationships can also be applied to surfaces at other places where AERONET stations are not present. Even at the AERONET sites, the surface reflectance relationships have large uncertainty that translates to uncertainty in the retrieved AOD. The bias correction algorithm proposed here can reduce that uncertainty in AOD. More importantly, it does not rely on AERONET surface and therefore can be

applied uniformly without assuming that the relationships derived over AERONET stations are valid everywhere.

As the reviewer requested, in the revised version of the paper, we added additional validation, analysis and radiative transfer studies on the bias correction approach to demonstrate the validity and applicability to AOD product from another retrieval algorithm, i.e. NASA's DT algorithm.

**In the rebuttal, authors suggest similar approaches have been reported but the examples provided are not convincing. The Lary et al(2009) study is one the first AI papers using MODIS aerosol observations and it is already more than 10 years old. Yet it has not resulted in corrections currently implemented anywhere in the MODIS (and other sensors to my knowledge) algorithms. The Gupta et al (2016) from the MODIS DT group restricts its applicability to very specific scenes where there are very clear and well-defined algorithm deficiencies. The corrections reported in Gupta et al (2016) study are only applicable to those conditions (urban areas). In contrast, the study under consideration here has a much larger scope (whole continental USA) with no much discussions of specifics of the scenes under considerations. In comparison with Gupta et al (2016), this study does not show the same level of detailed and specific analysis. Overall, in one case, the correction provided did not seem to have any meaningful impact or correction in the final product and in the other case, the corrections developed were well defined where they should be applied and it did result in a modified algorithm when those conditions happened.**

We respectfully disagree with the reviewer's opinion that our bias correction algorithm does not have a more general use. At NOAA, we always get requests from users for historic data and when we get a request, we provide the original ABI data along with bias corrected AODs. In fact, to compare the nature of our bias correction with the one published by Gupta et al., their approach is also ad hoc and in fact we find it internally inconsistent. The Gupta et al. approach uses MODIS surface reflectance product derived from a different land algorithm as a starting point. In that land surface reflectance algorithm, aerosol optical depth is also simultaneously derived using atmospheric correction algorithm where spectral surface reflectance ratios are prescribed. Therefore, the surface reflectance relationship fix derived in Gupta's approach depends on the relationships assumptions in the surface reflectance product and AOD derived. Unlike Gupta et al. algorithm that identifies urban areas based on certain criteria and applies a correction to AOD over those surfaces, our algorithm does not have to know a priori what the surface type is. It inherently identifies an error in AOD over surface types where the spectral surface reflectance relationships lead to errors.

Of course, one would like to improve the accuracy within the AOD retrieval algorithm itself and not in post processing. Such effort is under way, and it is waiting for implementation in routine runs so a sufficiently large sample could be obtained to evaluate its effectiveness.

**Overall, if the goal is a proposed methodology that it can be used elsewhere (other sensors, other algorithms) , this study can be of value even there is no addressing of the core surface reflectance problem. However, if this was the case, the authors should expand the methodology and demonstrate that it is usable in other settings/sensors/algorithms. This study as it is now is a well explained report on how an ad-hoc correction was derived and demonstrated to achieve its objective in correcting the differences observed.**

The application of the bias correction algorithm to NASA's dark target ABI AOD retrievals has been added in the revised paper. Section 5 describes the application of our bias correction to DT algorithm applied to GOES-16 ABI for the month of July 2019. The results show that the bias in the NASA GOES-16 ABI AOD is reduced when the bias correction algorithm is applied.

**Given the nature of this journal with its emphasis on operational and algorithmic atmospheric studies, I think this manuscript is border line with regards to publication. But ultimately, I think this should be an Editorial decision."**

**---**

**I also note that the Gupta et al (2016) paper mentioned has resulted in algorithmic updates to the algorithm in question there (MODIS Dark Target aerosols) – an algorithm change to the surface reflectance model, not a post-processing bias correction to the retrieval output (as in this manuscript).**

Our approach is more self-consistent because we are correcting the bias due to our own prescribed surface reflectance relationships. Gupta et al. approach is based on surface reflectance relationship from a separate land surface reflectance product. The MODIS surface reflectance product itself has high uncertainty in urban regions. In order to update the AOD algorithm with new spectral surface reflectance ratios, we have to first acquire a few years of data covering all seasons. This data gathering was especially difficult because the satellite was initially in a test position and then moved to its operational location. In its two different positions, the satellite was viewing different domains and the characteristics of these domains such as surface reflectance are different. It is easier to apply a correction post retrieval instead of waiting a long time to collect the data, analyze the data, derive new surface reflectance coefficients etc. Especially because unlike NASA, NOAA does not do frequent reprocessing; we have to provide users with the best operational product that we can generate.

In principle, the bias correction could be made part of the AOD retrieval algorithm, in which case it would become an algorithm change, applied to all pixels, that is all surfaces, not just urban. Also, in principle, the bias-corrected AOD could then be applied to derive a surface reflectance that would be consistent with the bias-corrected AOD and the satellite-observed reflectance. But since the purpose of the algorithm is AOD retrieval such a step would be unnecessary. It just goes to show that when the bias-correction improves the AOD it would also improve the surface reflectance.

Of course, one would like to improve the accuracy within the AOD retrieval algorithm itself and not in post processing. Such effort is under way, and it is waiting for implementation in routine runs so a sufficiently large sample could be obtained to evaluate its effectiveness.

**Line 430 says that the data processed using the correction algorithm are available from the author. But my impression from the paper is that this was being done in real-time as a correction to the operational product (i.e. one can get these retrievals now through the standard data server)? Is this not the case? If not, then my suggestion is to withdraw this paper and include some of this material in a follow-up improved algorithm paper; you note in your Response to Reviewers that "Improving the spectral surface reflectance relationships is the subject of an independent, parallel work, and thus it is not discussed in the current paper". However for me that the most interesting part is identifying the cause of the bias (which is partially discussed here) and fixing it (which is stated to be TBD). Either that or work on making the bias correction even better.**

**If this IS being implemented in the operational product (or will be within the coming month or so), a second path forward would be to expand the parts covering the existing surface model and showing why it is the main problem, as well as expanding the validation analysis with more (spatial) results to make this more informative for data users. I note that you added some additional**

**material to the revised manuscript, and appreciate those efforts, but think that more would be needed. Some comments and suggestions from me are below:**

We view the improvement of the current AOD algorithm by improving the surface reflectance estimation and application of the bias correction as two separate activities but having the same goal. The bias correction has the additional benefit that, in principle, the idea can be applied to any sensor and algorithm, while the improved ABI surface reflectance relationship is only applicable to ABI.

We will implement the bias correction algorithm soon and will provide the data in near-real-time on the NOAA ftp server. We think that even if we update/revise the spectral surface reflectance relationship in the algorithm, we may still need to apply the external bias correction to the derived AODs.

**The paper notes that the surface reflectance relationship is a function of NDVI and solar zenith angle, yet there is a clear solar angle dependent bias in the AOD retrievals at some of the sites shown in Figure 1. I went to the GOES-R ATBD cited as the source of these relationships and it says that they were empirical based on collocated ABI and AERONET data. However the ATBD does not show these relationships.**

The relationships are shown in p41 of ATBD (https://www.star.nesdis.noaa.gov/smcd/spb/aq/AerosolWatch/docs/GOES-R_ABI_AOD_ATBD_V4.2_20180214.pdf) and we also copied them in the response to reviewer #1. Following is the relationships and they are also included in the revised paper.

The surface reflectance relationships obtained are described in the following equations:

$$\rho_{0.47}[\rho_{0.64}] = (c_1 + c_2\theta_s) + (c_3 + c_4\theta_s)\rho_{2.25},$$

where $\rho_{0.47}$, $\rho_{0.64}$, $\rho_{2.25}$ are surface reflectance at the three bands, $c_1, c_2, c_3, c_4$ are coefficients depending on NDVI as shown in Table 3-12 of the ATBD (shown in the following), $\theta_s$ is the solar zenith angle. NDVI is defined by red (0.64 µm) and NIR (0.86 µm) bands at TOA as

$$\text{NDVI} = \frac{\rho_{0.86}^{TOA} - \rho_{0.64}^{TOA}}{\rho_{0.86}^{TOA} + \rho_{0.64}^{TOA}}.$$

Table 3-12. Coefficients in the spectral surface reflectance relationship for different ranges of NDVI.

| Channels (µm) | $c_1$ | $c_2$ | $c_3$ | $c_4$ |
|---|---|---|---|---|
| | $NDVI \geq 0.55$ | | | |
| 0.47 vs. 2.25 | 1.436330E-02 | 2.060893E-04 | 1.749239E-01 | -2.859502E-03 |
| 0.64 vs. 2.25 | 1.374160E-02 | -5.128175E-05 | 2.761044E-01 | 1.034823E-03 |
| | $0.3 \leq NDVI < 0.55$ | | | |
| 0.47 vs. 2.25 | 4.163894E-02 | -2.147513E-04 | 1.598440E-01 | 7.401292E-04 |
| 0.64 vs. 2.25 | 2.990101E-02 | -1.873911E-04 | 4.602174E-01 | 9.658934E-04 |
| | $0.2 \leq NDVI < 0.3$ | | | |
| 0.47 vs. 2.25 | 5.154307E-02 | 5.679386E-05 | 2.048702E-01 | -7.064656E-04 |
| 0.64 vs. 2.25 | 5.179930E-02 | -1.043257E-04 | 4.937035E-01 | 4.310074E-04 |
| | $NDVI < 0.2$ | | | |
| 0.47 vs. 2.25 | -4.990575E-02 | 2.138207E-03 | 8.498076E-01 | -1.179596E-02 |
| 0.64 vs. 2.25 | -3.397737E-02 | 1.640336E-03 | 1.087497E+00 | -9.538776E-03 |

**So one good way to expand the paper would be to show these relationships and then also look at whether the scatter around the relationships in the training data set shows systematic behaviour as a function of e.g. view zenith or scattering angle. Adding this sort of material could better support the conclusion that it is the surface model which is the main problem here. Figure 6 shows that the errors in AOD are a function of scattering angle but the paper is missing the link that it is surface that's the biggest component. I think we need to see the training data, and ideally retrieval simulations where a known surface error is introduced.**

We thank the reviewer for this suggestion. We have analyzed how errors in the parameterization (goodness of the empirical fit) can lead to AOD bias. First, training data used to derive the spectral surface reflectance relationship is collected at ABI-AERONET match-ups over full disk domain for the time period of 04/29/2017 – 01/15/2018 with the same cloud screening methods as those for high quality AOD retrievals to ensure clear sky pixels. Additional criteria are also applied to further filter the training data such as low AERONET AOD, spatial area around AERONET sites, etc. We probed why the bias varies with space and time, and in the paper used matchups with AERONET stations and regional AOD maps to demonstrate the problem. There are several reasons why the actual surface relationship can be different from the one obtained from the training set. (1) Individual pixels/AERONET sites/regions can be different in characteristics when compared to the whole data set. In the study, we use CONUS data instead of the full disk. We also showed analysis of an example at GSFC. For the data points shown in Figure 3 (a) and (b), the correlation between surface reflectances at 0.47 µm and 2.2 µm is very poor with a correlation coefficient of 0.4-0.7, depending on the date and time. The majority of points do not follow the general relationship derived using the full disk data. (2) The time period matters. We found that AODs over different time periods have different biases. For example, for July 2019, the original NOAA ABI high quality AOD has a bias of 0.06, which is similar in magnitude to the bias in the NASA's dark target AOD (in the revised paper), however, it is much higher than the bias of 0.01 in August-December 2018 (in the paper). In the paper, we also show three different days at GSFC as an example to demonstrate how different the real surface relationship is from the universal model at an individual site and time. (3) The training data used for deriving the surface reflectance relationships was constructed by using pixels from an area around the AERONET sites that is smaller than the area used in the validation, have low

AERONET AOD, and screened for the presence of clouds using criteria that essentially selects pixels with only high quality ABI AOD. Application of the surface reflectance relationship to other pixels, especially medium quality pixels, shows a degradation of ABI AOD against AERONET AOD.

We analyzed these limitations further and added a new section, Section 6 in the revised paper. We demonstrated that the patterns of the surface reflectance errors with respect to scattering angle in the training data set and in the data during the time period used in the paper over the full disk and the CONUS regions are different. We also show the magnitude of AOD errors introduced by the error in the surface reflectance through radiative transfer and AOD retrieval simulations. More importantly, the simulation results demonstrate that the bias due to surface reflectance error at 0.025 background AOD has the same magnitude as that at higher AOD, and that the bias obtained at 0.025 background AOD can be used to reduce the bias with higher AOD. Therefore, it proves that the AOD bias correction algorithm is equivalent to the reduction of surface reflectance bias.

**It also raises the question of why this relationship was used in the orginal algorithm? The study cites the MODIS Dark Target over-land AOD algorithm as a source of some assumptions; that algorithm parametrizes surface as a function of scattering angle rather than solar zenith angle. And we know that in the single-scatter limit, both aerosol and surface reflection are more directly linked to scattering angle than solar zenith. So some more background on the reason for choosing solar zenith angle would be useful (e.g. show the simulations indicating why this decision was made). Line 170 indicates that the training was done before the satellite was moved to its current position; however, again, to my knowledge this movement was planned and doing the calculating in terms of scattering angle may have meant that these coefficients would still remain useful for the new position.**

Surface spectral BRF relationships are complicated functions of four geometry parameters: solar zenith angle, solar azimuth angle, view zenith angle, and view azimuth angle. Using a single angle such as scattering angle or solar zenith angle won't completely model the relationship. The error in fitting the 0.47µm surface reflectance as a function of solar zenith angle in the training dataset shows little dependence on scattering angle. In addition, using data in one set of geometry to train the surface relationship and apply it in another set of geometry is a potential source of error. As an exercise to demonstrate the effect of the movement of the satellite from the test position ($89.5^{\circ}$ W) to the operational position ($75.2^{\circ}$W) on the surface reflectance relationship, a test was conducted to derive two surface relationship models: one using training data from October 2017 with GOES-16 in test position and another one using training data from October 2018 with the satellite at operational position. The two models are then applied to the October 2018 training data to estimate the surface reflectance at 0.47µm from that at 2.25 µm. The surface reflectance error (estimated – atmospheric corrected) at 0.47 µm band is plotted with respect to scattering angle and is shown in the figure below. It can be seen the patterns of the errors in the two data sets have some differences. The movement of the satellite may be one of the reasons that contributes to the differences. Other reasons can also cause the difference. For example, the sampling is not exactly the same in time and space for the above two time periods, i.e. a site may have more sampling pixels in one period than in the other due to the difference in atmospheric conditions, and vice versa.

[Figure]

*Figure A. Surface reflectance error at 0.47 µm band vs scattering angle of AERONET matchup data in October 2018 using model trained by October 2017 data and using model trained by October 2018 data.*

NASA's dark target algorithm uses scattering angle to do the modeling, but we can see that it also suffers from the same problem as shown in the revised paper.

We do mention that the change of geometry due to the change in satellite position is a potential source of error, but we don't say it is the main source of the error. The main problem is that the individual pixels/regions/time periods can have very different behavior than the model derived from the whole full disk training data set. Therefore, we find it appropriate and important to externally fix the bias at every pixel.

**The paper mentions MODIS and VIIRS AOD algorithms a lot (as there is some commonality between them and ABI). Yhis is mostly talking about NASA MODIS Dark Target and NOAA VIIRS (as opposed to e.g. NASA VIIRS Dark Target and Deep Blue). This should be made clearer. Maybe we need a statement early on saying that references to VIIRS mean NOAA algorithms.**

Added NOAA VIIRS AOD wherever we mention VIIRS AOD.

**Line 53: MODIS Atmospheres Collection 6 onwards is both 3 km and 10 km. This is also a place where it should be specified that it is NOAA VIIRS data, as NASA VIIRS is 6 km.**

Revised to "The NOAA ABI AOD product has a spatial resolution of 2 km at nadir, compared to 3 km and 10 km from MODIS Collection 6 and 750 m (NOAA product) and 6 km (NASA product) from VIIRS."

**Line 122: this is a bit misleading: AERONET provides a subset of those wavelengths (dependent on instrument), not all of them.**

Revised to "… it is measured at a subset of 22 different wavelengths from ultraviolet to infrared, i.e. 340, 380, 400, 412, 440, 443, 490, 500, 510, 531, 532, 551, 555, 560, 620, 667, 675, 779, 865, 870, 1020, and 1640 nm, depending on the specific instrument."

**Line 127: I understand there is a delay of up to a year or so between AERONET level 1.5 and level 2 data. However it seems that level 2 should be available by now for the study period, so these ideally should be used.**

We have pointed out in the previous response to reviewer #1 and rechecked the status on August 8, 2020. Many sites still don't have Level 2 data. We have been using GSFC site's data for analysis extensively. As of August 8 2020, GSFC still does not have level 2 data available for the days after September 2018. In an email from Brent Holben at GSFC site in April 2020 when we asked him about the level 2 data at GSFC, he said that "*I just talked to Tom Eck our Langley Calibration guy who manages the MLO and GSFC instruments. The current instrument has not been recalibrated since Sept 2018 but has been compared to other master instruments freshly calibrated at MLO. Each time the current master always looked as good or superior spectrally to the freshly calibrated instruments so he has not returned the current instrument to MLO for a final cal. The bottom line is the level 1.5 from the GSFC instrument is calibrated better than probably any instrument in the network, it just hasn't been thru the protocols to raise it to level 2.*" So, we don't think it is wise to discard the data just because it is level 1.5.

In our daily work, we routinely do our analysis with both Level 2 and Level 1.5 data and we find the Level 1.5 data to be suitable for our analysis.

A scan of the dataset shows that only 64 out of 80 sites have updated level 2 data beyond December 2018. And only 40 out of 77 sites have updated level 2 data beyond July 2019. In the paper, we use the time period August-December 2018 for NOAA ABI AOD and July 2019 data for NASA dark target ABI AOD analysis (added in the revised version).

As seen in the reviewer's comment below, level 1.5 has a bias of up to 0.02 and an uncertainty of 0.02. We don't think we could gain much by switching to level 2 data, but we would lose 20% - 48 % of sites if we did it, and would also lose the analysis at GSFC site.

**I am also not sure about the statement that level 1.5 has a "bias" of 0.02. It looks like this was taken from the Abstract of the Giles paper cited here. But the relevant passage of that paper is more nuanced: "Therefore, the quality of the Level 1.5 near-real-time AOD changes with time with high-quality data at the start of the deployment but up to a +0.02 bias and 0.02 uncertainty for data collected more than 1.5 years since pre-field calibration." The conclusions to that paper also note "up to" 0.02. I am not sure why the Abstract to that paper omits "up to".**

Added "up to" in the text.

**Section 4: As I understand it, the bias correction is representing the AOD error as two quadratic functions of time of day (split by whether the Sun is east or west of the sensor), fit based on the difference between retrieved AOD and that from a 30-day minimum retrieved AOD (on a timestep**

**basis), also subtracting a background AOD from AERONET (taken as 0.025). The flow chart here is useful but I think it would be good to have an example showing actual data, perhaps for one of the cases shown in Figure 1.**

A Figure (Figure 5 in the revised paper) is added for a pixel close to GSFC to illustrate the 30-day clear AOD composite process.

**Since it is known that negative AOD is unphysical (but permitted in the retrieval), wouldn't it make sense for the bias correction here to also set negative AOD to zero? My understanding is that this is an inherited assumption from MODIS Dark Target retrievals which causes a few issues and misunderstandings for some users.**

We are just following the convention and do not find any problem with that. NOAA VIIRS AOD, ABI AOD and NASA dark target AOD all use -0.05 as minimum. We would like to keep it to be consistent with our existing products.

**Line 277: If the retrieval is being done every 5 minutes, I don't think it makes sense to average the AERONET data in an hour around the time. If there is real AERONET variation, it would be smoothed out by this averaging. Why not compare with the AERONET measurements directly associated with each time step? I understand this choice was the same as the done for earlier VIIRS analyses, but as pointed out in this paper, polar and geostationary are quite different sampling types. Or are the ABI retrievals also being averaged to hour time steps? This was not clear. It would be better to match up without distorting the sampling too much.**

Again, the use of this matchup criteria has become a standard and is also used on matchups between AERONET and geostationary data. For example, in a recent paper by Gupta et al. (2019) on AHI AOD retrieval, they used similar criteria and stated that "Thus, the temporal mean AOD of all AERONET AOD measurements within 30 min of an AHI scan will be compared with the spatial mean of all Level 2 AHI-retrieved AOD values within a 0.25°x0.25° box centered at the AERONET station. This method of matching spatiotemporal statistics, in one form or another, has become a standard within the aerosol remote sensing community (Levy et al., 2010; Petrenko et al., 2012; Remer et al., 2013; Huang et al., 2016; Gupta et al., 2018)." A standard like this helps the comparison of accuracy and precision values reported by different research groups using different AOD retrieval algorithms. However, we agree that determining the best way to compare satellite and ground AODs may need further investigation. Such an investigation must be done before we switch to other criteria, e.g. changing the size of time window and spatial averaging area. For now, our purpose is to demonstrate the effect of the bias correction algorithm.

**Line 356: Wallops may be rural but the 27.5 km circle around it includes a large amount of coastal areas, as well as small towns, so I am not sure I agree fully with the statement about favourability.**

Changed to "Wallops is a site with mixed pixels of rural, small town and water,…" Coastal pixels are screened in the AOD retrieval algorithm and therefore are not included in the statistics.

**Section 5: it would be good to have some maps of site-specific validation metrics for the "before" and "bias-corrected" cases, as well as some maps (maybe a seasonal composite?) of ABI AOD for certain times of the day. Continental all-sites metrics hide this information. This would give the reader a sense of how big the differences are, and how much performance has improved. While the examples shown are useful, it's hard to know how representative they are, or how much retrievals change in those parts of CONUS away from AERONET sites.**

We added maps of correlation coefficients, mean bias and RMSE.  We also added maps of monthly mean AOD for September 2018 as an example to show the mean AOD at three time steps: 1500 UTC, 1700 UTC and 2000 UTC.

**Figures 1, 8: the presentation here needs to be improved – at least site names should be shown on panels, not only in the Figure 1 caption.**

Added.

**Figure 5: the regression fits should be removed from this Figure, as these data violate the assumptions required for ordinary least squares to give unbiased, robust results. I think the 1:1 line and red points get the main message across anyway.**

Removed.

**Figure 12: I don't think the brief PM bit really fits here. Yes, the correlation is increased but we don't always expect them to be correlated. The regression comments from earlier also apply here. I suggest removing this. If the authors want to note that better AOD can help downstream derived products like air quality forecasting, I think it's fine to just say that.**

We respectfully disagree and would like to retain this figure.  One of the main reasons we developed the bias correction algorithm for AOD is because of the need to derive PM2.5 for operational air quality monitoring applications.  We understand that in addition to AOD, the relationship between AOD and PM2.5 depends on many other parameters such as relative humidity, boundary layer height, aerosol type etc.  Here, we are showing that having an accurate AOD is important for PM2.5 estimates.  Given that the only difference between the two figures (before and after AOD bias correction) is improved AOD, the results show how correlation can be improved if AOD accuracy is improved.

**Reference**

Gupta, P., Levy, R. C., Mattoo, S., Remer, L. A., Holz, R. E., and Heidinger, A. K.: Applying the Dark Target aerosol algorithm with Advanced Himawari Imager observations during the KORUS-AQ field campaign, Atmos. Meas. Tech., 12, 6557–6577, https://doi.org/10.5194/amt-12-6557-2019, 2019.

---

## Author Response (AR3)

Dear Andrew,

Thank you for the review of the paper. Given below are our responses (normal font) to individual reviewer comments (bold font).

**1. Line 79: I suggest moving the formal NDVI definition from the new section 6 to here, since this is the first time NDVI is discussed in the context of the ABI algorithm.**

Moved.

**2. Line 131: "Angstrom" should read "Ångström".**

Changed.

**3. Line 171: I think this should be "in the surface reflectance model" rather than "in surface reflectance retrievals"**

Changed.

**4. Line 236: I would delete "the" in front of "reflectance space" and "AOD space".**

Deleted.

**5. Line 291: this section is quite long and would be more readable if some subsection breaks were added. Perhaps these could be "5.1 Application to NOAA ABI data", "5.2 Application to Dark Target ABI data", "5.3 Impact on particulate matter estimation" or similar?**

Subsection breaks added.

**6. Line 370: Figure 10 stands alone and feels a bit odd as a paragraph in between discussions of specific AERONET sites. I suggest moving this text and Figure slightly earlier, to line 350 where Figure 9 is currently introduced. This means that Figures 9 and 10 will switch numbers in the paper. I think that would fit better because you then go from analyzing results in aggregate (e.g. overall stats, error vs. angle and vs. AOD, consistency of times of day) to site-specific discussions.**

The two figures and discussions are switched.

**7. Line 443: I would add a citation to a Dark Target geostationary paper again here, e.g. Gupta (2019) as used before, for the reader's convenience.**

Added**.**

**8. Section 6: I appreciate this new material although think it could be organized a bit more clearly. I suggest creating subheadings 6.1 and 6.2, detailing (1) the coefficient fitting and (2) the use of 6SV to generate and analyze the radiative transfer simulated retrievals.**

The section is divided into two subsections: 6.1 Surface reflectance model bias analysis, and 6.2 Radiative transfer simulation analysis.

**9. Line 499: the notation p0.47[p0.64] is not ideal here: I understand what you mean, but a less familiar reader might not, and the notation used mentioning both wavelengths is nonstandard. I think this would be better writing separate equations for each of p0.47 and p0.64, and likewise defining coefficients c1-c8.**

The equations are separated.

**10. Line 504: Should this say the "coefficients" rather than the "equations" are obtained?**

Changed**.**

**11. Line 543: I recommend replacing the word "prove" with "show" because I feel "prove" is too strong (and you do not prove it in a mathematical sense here, you show it empirically).**

Replaced**.**

**12. Figure 7: I suggest adding a horizontal line along AOD=0 here (similar to how it is in Figure 8), which will help the reader see the overall sign and magnitude of bias.**

Added**.**

**13. Figure 9: there is a site in northern Texas (I believe it is NEON_CLBJ) where the bias correction decreases correlation, increases bias, and makes little change to RMSE. This site's description indicates it is surrounded by mixed forest and grassland (https://aeronet.gsfc.nasa.gov/new_web/photo_db_v3/NEON_CLBJ.html). I recommend looking into what happened here and see if some explanation is forthcoming. I'm not sure, for example, if retrievals did get worse or there is simply a low data volume here and there is therefore a large uncertainty on these summary statistical metrics. If not already done, sites with a low number of matchups might be excluded from the analysis. I noted a similar degradation in the site in Utah as well.**

We noticed an error in our code for generating these figures when examining this site. After correcting the error and replotting the figures, the NEON_CLBJ site looks fine with a 0.01 increase in bias and not much change in correlation and RMSE. But we do notice that some sites have decreased AOD accuracy. After examination, we added following paragraph to discuss this issue in the paper. When generating the figures, we remove the sites with low matchup volume using a threshold of 400 matchups.

*Overcorrection, under-correction and/or decrease of correlation are observed at several sites. For example, at NEON_TALL (32.950°N, 87.393°W) in AL, the correlation coefficient decreases from 0.88 to 0.78, the bias decreases from 0.01 to 0, and RMSE remains 0.06. In the bias correction algorithm, AOD is assumed to hit the background AOD of 0.025 at least once during 30-day period for most of the time steps in order to generate correct curve for bias correction. If this assumption is not satisfied, the algorithm's performance will decrease. If the lowest AOD is higher than the background AOD during the 30-day time period for a pixel for some or all of the time steps, the derived AOD bias curve will be distorted and overcorrection will occur for those time steps. Similarly, slight under-corrections may occur if the lowest AOD during the 30-day period is lower than 0.025.*

**14. Figure 10: I notice the AOD changes between left and right not only over land, but over water as well. Was the same bias correction applied over water? This would seem inappropriate, since the discussion about the approach is restricted to land scenes, and the retrieval approach and surface reflectance uncertainties are different over water. This was not discussed in the text, as far as I can tell. Further, there was no discussion of whether the water retrievals suffered from biases, or analysis of whether the uncertainties were improved using the bias correction. If application over water was an error, then I would suggest updating this figure, or possibly masking out the water locations to avoid confusing the reader. If the application over water was intentional then this really needs to be discussed (and justified and analyzed in the same way) in more detail in the paper.**

The figure was replotted and the water pixels were masked.

Hai and co-authors

[revised manuscript text omitted]